# Valid Inference with Imperfect Synthetic Data

**Yewon Byun**[1], **Shantanu Gupta**[1], **Zachary C. Lipton**[1], **Rachel Leah Childers**[2], **Bryan Wilder**[1]

Machine Learning Department, Carnegie Mellon University[1]     University of Zurich[2]

{yewonb, shantang, zlipton, bwilder}@cs.cmu.edu, rachel.childers@df.uzh.ch

## Abstract

Predictions and generations from large language models are increasingly being explored as an aid in limited data regimes, such as in computational social science and human subjects research. While prior technical work has mainly explored the potential to use model-predicted labels for unlabeled data in a principled manner, there is increasing interest in using large language models to generate entirely new synthetic samples (e.g., synthetic simulations), such as in responses to surveys. However, it remains unclear by what means practitioners can combine such data with real data and yet produce statistically valid conclusions upon them. In this paper, we introduce a new estimator based on generalized method of moments, providing a hyperparameter-free solution with strong theoretical guarantees to address this challenge. Intriguingly, we find that interactions between the moment residuals of synthetic data and those of real data (i.e., when they are predictive of each other) can greatly improve estimates of the target parameter. We validate the finite-sample performance of our estimator across different tasks in computational social science applications, demonstrating large empirical gains.

## 1 Introduction

Practitioners increasingly leverage large language models (LLMs) as cheap but noisy labelers for automating tasks traditionally reliant on manual human annotations Ziems et al. [2024]. Beyond annotation, recently, practitioners have started to explore the possibility of leveraging LLMs for more diverse and open-ended forms of model-generated data, such as outputting entirely new synthetic samples, e.g., simulating human responses to surveys or human participants in early pilot studies [Argyle et al., 2023, Brand et al., 2023, Dominguez-Olmedo et al., 2024, Anthis et al., 2025, Hwang et al., 2025b]. Determining the extent to which researchers should integrate LLM simulations—whether by simulating all samples, combining simulated and real samples, or relying entirely on human participants—remains an open and task-dependent question. While such pipelines leveraging fully synthetic simulations have yet to be fully realized, reliable mechanisms for aggregating these data sources are indeed what will inform both the feasibility of such design choices and how such pipelines should be implemented in practice.

A persistent challenge, however, is that naively aggregating synthetic samples with real data for downstream inference often leads to greatly biased estimates, compromising the statistical validity of downstream conclusions. Ideally, we would like to realize the benefits of incorporating information from these additional data sources while retaining favorable statistical properties—consistency and proper asymptotic coverage. We consider the setting where practitioners have access to a corpus of unlabeled text and a small set of human-annotated samples with labeled covariates and outcomes. Here, practitioners can leverage LLMs to (1) predict covariates and outcomes for the unlabeled text samples; and (2) generate new text samples conditioned on available samples and label the covariates and outcomes for them similarly to (1).

First of all, it is not immediately obvious how to even produce synthetic samples such that they can be used in a principled manner. Naively drawing samples from a generative model and treating them

39th Conference on Neural Information Processing Systems (NeurIPS 2025).

as additional samples alongside real data makes it impossible to provide statistical guarantees for the resulting estimate if the generative model does not perfectly match the real distribution, which is expected in practice. We propose a specific sampling strategy in which each synthetic sample is generated conditional on an individual real text in Section 3. What makes this formulation statistically powerful is that it introduces a correlation structure between each real text and synthetic sample. We will leverage this correlation structure, as it enables us to more effectively share information across real and synthetic data.

In Section 4, we introduce a new estimation framework based on Generalized Method of Moments (GMM). The GMM framework allows us to incorporate multiple sources of information by adding moments. The optimal weighting in GMM produces a combination of these moments that minimizes the variance of all estimators based on these moments [Chamberlain, 1987]. This optimal weighting measures the cross-correlations between the synthetic and real data, producing a weighting matrix that reduces the variance of the real data moments if there is information from the synthetic data moments. Prospectively, it is not intuitive that the incorporation of additional moments based exclusively on synthetic data (defined in terms of a separate parameter from the target parameter) should yield any benefits (or *even affect*) the estimation of the target parameter of the real data. Intriguingly, we find that the incorporation of synthetic data leads to better estimation and tighter confidence intervals when the synthetic data moment residuals are predictive of the real data moment residuals. When they are independent from each other, the variance reduces to the optimal variance based only on the fully observed data. That is, in the worst case where synthetic data is *completely uninformative*, including it does not hurt (at least asymptotically). Finally, in Section 5, we analyze the finite-sample performance of our estimator using real-world datasets that encompass varying computational social science tasks, demonstrating large empirical gains.

At a fundamental level, this work takes a step towards understanding how synthetic data from foundation models can systematically be leveraged to support valid inference. As the usage and future promise of foundation models continue to grow, so too will the complexity of pipelines that incorporate their outputs. Our framework provides a foundation for easily extensible estimation methods that can safely incorporate the growing variety and quality of synthetic data sources from such models. More broadly, this GMM-based estimation framework for incorporating auxiliary data may be of broader interest as an alternative to the predominant debiasing-based methods in the surrogacy literature [Angelopoulos et al., 2023a], as it can more flexibly accommodate multiple proxy covariates and proxy outcomes compared to existing approaches.

## 2    Related Work

**Statistical Inference and Debiasing Methods.**    Our work is broadly related to performing statistical inference with missing data, where past works have explored approaches to yielding valid and efficient parameter estimates [Robins et al., 1994]. Other work has notably explored the usage of ML models to estimate nuisance parameters [Chernozhukov et al., 2018]. The most related line of research are debiasing methods that focus on combining ground truth data with surrogate predictions (often produced by a machine learning model) to perform statistical inference [Egami et al., 2023, Gligorić et al., 2024]. These frameworks are often referred to as prediction-powered inference [Angelopoulos et al., 2023a,b] in the machine learning literature. Such methods have been well-studied in the context of predicted outcomes and, more recently, predicted covariates [Ji et al., 2025]. A key difference between these works and our setting is that the primary focus of our work is how to incorporate fully synthetic samples, which remains unaddressed by previous work. For clarity, we refer to samples as fully synthetic when (1) the underlying text is synthetically generated and (2) both its covariates and outcomes are model predictions.

**LLMs for Data Annotation and Synthetic Simulation Tasks.**    Our work is motivated by the increasingly growing use and future promise of LLMs for annotations and simulations [Ziems et al., 2024, Hwang et al., 2025a, Anthis et al., 2025]. There has been growing interest in using LLMs in fully synthetic simulation studies, with primary applications in exploratory research or early pilot studies. For instance, recent work has studied simulating social interactions and behaviors [Chen et al., Park et al., 2023]. Other works have explored LLMs for simulating survey responses [Dillion et al., 2023, Rothschild et al., 2024, Dominguez-Olmedo et al., 2024], analyzing how well simulations approximate human responses while cautioning about drawbacks such as limited diversity and lack of context-awareness. In summary, this line of work shows the potential of incorporating synthetic

data powered through strong generative models in such downstream pipelines but also exhibits clear failure modes and imperfect conclusions from such studies. While most of these works focus on qualitative takeaways and early signals for future experiments, we focus on a forward-looking setting of making statistically valid inference given such synthetic samples.

## 3 Preliminaries

**Notation and Setup.** We consider a parameter estimation task where the goal is to estimate a target parameter $\theta^\star \in \mathbb{R}^d$. Let $(T, X, Y) \sim \mathcal{D}$ denote a random triple drawn from an unknown data-generating distribution $\mathcal{D}$ over text inputs $T \in \mathcal{T}$, covariates about the text (e.g., structured metadata) $X \in \mathcal{X} \subseteq \mathbb{R}^d$, and labels $Y \in \mathcal{Y}$. For example, $T$ can be texts from online requests, where $X$ are linguistic markers of hedging (i.e., notions of uncertainty) and $Y$ is perceived politeness. Due to labeling budget constraints, we assume we only observe a small fraction of human-annotated data (i.e., ground-truth covariates and labels about the text). Specifically, we have access to labeled dataset $\mathcal{D}_{\text{labeled}} = \{(T_i, X_i, Y_i)\}_{i=1}^n$ that is sampled i.i.d. from $\mathcal{D}$ and an unlabeled corpus of text $\mathcal{D}_{\text{unlabeled}} = \{(T_j)\}_{j=n+1}^{n+m}$ sampled i.i.d. from $\mathcal{D}_T$ (i.e., the marginal distribution over $T$), where $m \gg n$. To supplement this limited supervision, we leverage machine learning models in the following two ways.

**Proxy Covariates and Labels.** We use a machine learning model $f$ to produce predictions $\{f_X(T_j), f_Y(T_j)\}$ for the available set of input texts $T \in \mathcal{T}$. Here, $f_X$ and $f_Y$ denote the same machine learning model, using separate prompts for the target outcome (either a covariate $X$ or outcome $Y$) (see Appendix E for details). This yields the following $\mathcal{D}_{\text{proxy}} = \{(T_i, f_X(T_i), f_Y(T_i)\}_{i=1}^n \cup \{(T_j, f_X(T_j), f_Y(T_j)\}_{j=n+1}^{n+m}$. For simplicity, we will refer to these as **proxy samples** and denote them as $(T, \hat{X}, \hat{Y})$. We will refer to the distribution over proxy samples as $\hat{\mathcal{D}}$. This is the main setting generally considered in the surrogacy literature (restricted to predicted outcomes).

**Synthetic Covariates and Labels.** We propose a new data augmentation process which generates new samples using the same machine learning model $f$ (employing it as a generative model, instead of a classifier). Specifically, our method conditions the generation process on each individual text $T_j$ as an example and asks the model to generate a new synthetic sample given that context. Formally, for each $i$, we sample a new text $\tilde{T}_i$, conditioned on $(T_i, X_i)$ if the sample is labeled and $(T_j, \hat{X}_j)$ if the sample is unlabeled (since $X_j$ is not available, by definition). More concretely, $\tilde{T}_k \sim \mathbb{P}(\cdot \mid T_i, X_i)$ if labeled and $\tilde{T}_k \sim \mathbb{P}(\cdot \mid T_j, \hat{X}_j)$ if unlabeled. See Appendix E for prompts used for synthetic data generation. Based on the generated sample, which we denote as $\tilde{T}_i$, we then extract its corresponding covariates and outcomes using $f$ similarly as in proxy samples. More concretely, $\tilde{X}_k \sim \mathbb{P}(\cdot \mid \tilde{T}_k)$ and $\tilde{Y}_k \sim \mathbb{P}(\cdot \mid \tilde{T}_k)$. This results in $\mathcal{D}_{\text{synthetic}} = \{(\tilde{T}_k, \tilde{X}_k, \tilde{Y}_k)\}_{k=1}^{n+m}$. We refer to the distribution over **synthetic samples** $(\tilde{T}, \tilde{X}, \tilde{Y})$ as $\tilde{\mathcal{D}}$.

This specific sampling process has two motivations. First, from a machine learning perspective it can be seen as a form of in-context prompting, where the model is given an example from the dataset in order to align it more closely with the task. Iteratively prompting with different samples $T_i$ is also likely to produce more diverse samples than asking for many samples with the same prompt. Second, from a statistical perspective, it introduces a correlation structure between each real text $T_i$ and synthetic sample $\tilde{T}_i$.

Finally, we introduce some notation that combines all of these data sources into draws from a single joint distribution. Specifically, we introduce a new random variable $s \in \{0, 1\}$ which is an indicator for whether $T$ is labeled (1) or unlabeled (0). Then, we view the complete generative process as draws $(T, s, s \cdot X, s \cdot Y, \tilde{X}^1, \tilde{Y}^1 ... \tilde{X}^M, \tilde{Y}^M)$ for $M$ different kinds of auxiliary data. So far, we have discussed two kinds, proxy and synthetic, that we employ empirically ($M = 2$), but our methods are fully extensible to additional kinds of auxiliary data. For example, we could include samples from multiple different generative models. The real $(X, Y)$ are observed only for labeled points with $s = 1$ while the auxiliary data is available for all samples. The joint distribution over this full tuple is induced by the composition of the generative processes for the components described above.

# 4 Combining Synthetic Information via Generalized Method of Moments

To estimate the target parameter $\theta^\star$, we adopt a generalized method of moments (GMM) approach [Hansen, 1982] that combines information from the different types of data in the following manner.

## 4.1 Moment Conditions

Our framework is applicable whenever the target parameter can be identified by a set of moment conditions, functions whose expectation should be zero at the true value of the parameter. Moment-based estimation is a broad and flexible framework that includes almost all commonly used statistical frameworks (e.g., maximum likelihood, generalized linear models, instrumental variables, etc). We begin by defining the moment conditions that identify $\theta^*$ under the distribution of interest (i.e., the real-data distribution $\mathcal{D}$). In the following section, we introduce how this can be adapted to incorporate surrogate data (i.e., proxy and synthetic data).

Formally, we consider the problem of estimating a parameter $\theta \in \mathbb{R}^d$. The true value $\theta^*$ is identified as the solution to a set of $p \geq d$ moment conditions

$$\mathbb{E}[\psi^{(\ell)}(\theta^*)] = 0, \quad \ell = 1...p$$

where the $\psi^{(\ell)}$ are continuously differentiable functions $\mathbb{R}^d \rightarrow \mathbb{R}$. For example, in a maximum likelihood model, we would have one $\psi$ for the derivative of the log-likelihood with respect to each parameter, and the moment conditions enforce that $\theta^*$ satisfies the first-order conditions for maximizing the likelihood. Let $\psi(\theta) = [\psi^{(1)}(\theta)...\psi^{(p)}(\theta)]^\top$ denote a column vector stacking the $p$ moments.

## 4.2 Constructing Our GMM Estimator

To leverage the auxiliary data (i.e., proxy data and synthetic data) in making our GMM estimator more efficient, we can construct a set of auxiliary moments for each additional source of data. We estimate an additional set of auxiliary parameters $\eta_1, ..., \eta_M \in \mathbb{R}^p$, one parameter vector for each set of new auxiliary data. In the specific instantiation of the model that we use here, we always have $M = 2$ (proxy and synthetic data), but in principle our method is extensible to many sources of auxiliary data, for example synthetic samples generated from several different models. Roughly, each new parameter vector $\eta_i$ can be understood as the parameter that we would estimate using each auxiliary data source, and our augmented model will automatically determine how to use these auxiliary estimates to inform the estimate of the parameter of interest $\theta$.

For each new parameter vector $\eta_i$, we introduce a corresponding set of new moments to estimate this parameter and allow its estimate to inform the estimate of $\theta$. Specifically, we introduce for each $\eta_i$ two new blocks of moments that are copies of the original moments for $\theta$. Intuitively, one block of moments will be evaluated only on the real (labeled) data, while the other will be taken on the pooled set of labeled data and auxiliary dataset $i$. The pooled-data moment will allow us to improve the estimation of $\eta_i$ using the larger sample. The version evaluated only on the real data will allow GMM to evaluate how well the moments for the auxiliary parameter correlate with those of the true parameter on the same data, and share information across them if the auxiliary moments are informative (as we would expect if the generated data is high quality).

Formally, let $S_t \in \mathbb{R}^p$ stack $p$ copies of the indicator variable $s_t$ for whether a data point $t$ is labeled. In block matrix notation, the combined model takes the form of the augmented moments

$$g_t(\theta, \eta) = \begin{bmatrix} S_t \\ S_t \\ \vdots \\ S_t \\ 1 \\ \vdots \\ 1 \end{bmatrix} \odot \begin{bmatrix} \psi(\theta) \\ \psi(\eta_1) \\ \vdots \\ \psi(\eta_M) \\ \psi(\eta_1) \\ \vdots \\ \psi(\eta_M) \end{bmatrix} \in \mathbb{R}^{p+2Mp} \tag{1}$$

We will then jointly estimate $(\theta, \eta)$ as the solution to the moment condition $\mathbb{E}[g_t(\theta, \eta)] = 0$. For clarity, we refer to our estimator that uses real and proxy data ($M = 1$) as **GMM-Proxy** and our estimator that uses real, proxy, and synthetic data ($M = 2$) as **GMM-Synth** throughout the paper. We remark that since the parameter of interest $\theta$ appears only in its original set of moments, which are evaluated only on the labeled data, this new moment condition still identifies the target parameter $\theta^*$. However, as we discuss below, when we apply standard methods for efficiently estimating the augmented GMM, the new moment conditions will be leveraged to reduce the variance of the estimate without compromising consistency or asymptotic normality.

Before turning to estimation, we provide a concrete example of our moment construction for the case of generalized linear models (GLMs) in two-dimensions.

### 4.3 Example 1. Generalized Linear Models

Recall that the standard GLM formulation optimizes the objective function,

$$\ell_\theta(x, y) = -yx^T\theta + f(x^T\theta),$$

where $f$ is a function that is convex and smooth. We remark that this recovers the setting of logistic regression when $f(z) = \log(1 + \exp(z))$. Let us assume a two-dimensional setting for illustration. This translates to the population moment conditions of

$$\mathbb{E}\left[X_1 Y - \frac{\partial f(X^T\theta^*)}{\partial \theta_1}\right] = 0, \quad \mathbb{E}\left[X_2 Y - \frac{\partial f(X^T\theta^*)}{\partial \theta_2}\right] = 0$$

We have similar moments for proxy and synthetic data, where we use parameters $\eta = (\eta^{(1)}, \eta^{(2)})$, which are also two-dimensional. Within our GMM framework, we construct the following set of moment conditions across the observed, proxy, and synthetic data.

$$g_t(\theta, \eta) = \begin{bmatrix} s_t \\ s_t \\ s_t \\ s_t \\ s_t \\ s_t \\ s_t \\ 1 \\ 1 \\ 1 \\ 1 \end{bmatrix} \odot \begin{bmatrix} X_{t,1}Y_t - \frac{\partial f(X_t^T\theta)}{\partial \theta_1} \\ X_{t,2}Y_t - \frac{\partial f(X_t^T\theta)}{\partial \theta_2} \\ \hat{X}_{t,1}\hat{Y}_t - \frac{\partial f(\hat{X}_t^T\eta^{(1)})}{\partial \eta_1^{(1)}} \\ \hat{X}_{t,2}\hat{Y}_t - \frac{\partial f(\hat{X}_t^T\eta^{(1)})}{\partial \eta_2^{(1)}} \\ \tilde{X}_{t,1}\tilde{Y}_t - \frac{\partial f(\tilde{X}_t^T\eta^{(2)})}{\partial \eta_1^{(2)}} \\ \tilde{X}_{t,2}\tilde{Y}_t - \frac{\partial f(\tilde{X}_t^T\eta^{(2)})}{\partial \eta_2^{(2)}} \\ \hat{X}_{t,1}\hat{Y}_t - \frac{\partial f(\hat{X}_t^T\eta^{(1)})}{\partial \eta_1^{(1)}} \\ \hat{X}_{t,2}\hat{Y}_t - \frac{\partial f(\hat{X}_t^T\eta^{(1)})}{\partial \eta_2^{(1)}} \\ \tilde{X}_{t,1}\tilde{Y}_t - \frac{\partial f(\tilde{X}_t^T\eta^{(2)})}{\partial \eta_1^{(2)}} \\ \tilde{X}_{t,2}\tilde{Y}_t - \frac{\partial f(\tilde{X}_t^T\eta^{(2)})}{\partial \eta_2^{(2)}} \end{bmatrix}$$

### 4.4 GMM Estimation

Given our augmented moment conditions $g$, we estimate the parameters $(\theta, \eta)$ by minimizing the GMM objective:

$$\hat{\theta}_T, \hat{\eta}_T = \arg \min_{\theta \in \Theta, \eta \in \mathbb{R}^{2Mp}} \widehat{Q}_T(\theta, \eta), \tag{2}$$

where

$$\widehat{Q}_T(\theta, \eta) = \left[\frac{1}{T}\sum_{t=1}^{T} g_t(\theta, \eta)\right]^\top \widehat{\mathbf{W}}_T \left[\frac{1}{T}\sum_{t=1}^{T} g_t(\theta, \eta)\right]. \tag{3}$$

**Two-step GMM estimator.** We adopt the two-step GMM procedure as described in Newey and McFadden [1994]. First, we compute the one-step estimator $\hat{\theta}_T^{(\text{os})}, \hat{\eta}_T^{(\text{os})}$ using an identity weight matrix $\widehat{\mathbf{W}}_T = \mathbf{I}$. Then, we estimate the optimal weight matrix as:

$$\widehat{\Omega}_T(\hat{\theta}_T^{(\text{os})}, \hat{\eta}_T^{(\text{os})}) = \left[\frac{1}{T} \sum_{t=1}^{T} g_t(\hat{\theta}_T^{(\text{os})}, \hat{\eta}_T^{(\text{os})}) g_t(\hat{\theta}_T^{(\text{os})}, \hat{\eta}_T^{(\text{os})})^\top\right], \tag{4}$$

and set

$$\widehat{\mathbf{W}}_T = \left[\widehat{\Omega}_T(\hat{\theta}_T^{(\text{os})}, \hat{\eta}_T^{(\text{os})})\right]^{-1}. \tag{5}$$

This optimal weighting has the interpretation as the inverse empirical covariance of the moment conditions on the one-step estimate. We then compute the final two-step estimator by minimizing $\widehat{Q}_T(\theta)$ with this updated weighting matrix. This choice of $\widehat{\mathbf{W}}_T$ yields an asymptotically efficient estimator under standard GMM regularity conditions. Following Chamberlain [1987], this choice of weighting minimizes the semiparametric efficiency bound with respect to the semi-parametric model defined by these moments (see Appendix B for more details).

The adoption of two-step GMM is a critical component of our proposed estimation framework. Indeed, in the first-step estimates, the synthetic and proxy data will have no impact on the estimate of $\theta$ because they never appear in the moment conditions concerning $\theta$. In the second stage though, the weight matrix $\widehat{\mathbf{W}}_T$ accounts for the covariance between moment conditions, where off-diagonal terms in the matrix allow moments for the auxiliary data sources to influence the estimation of $\theta$.

### 4.5 Consistency and Asymptotic Inference

We now present results on the consistency and asymptotic behavior of our GMM estimators.

**Proposition 1.** *Our estimate $\hat{\theta}_T$ (as defined in Equation 2) is consistent and asymptotically normal. It converges in distribution as*

$$\sqrt{T}((\hat{\theta}_T', \hat{\eta}_T')' - (\theta', \eta')') \xrightarrow{d} \mathcal{N}(0, V)$$

*where the covariance $V$ is given by*

$$V = \left(G(\theta, \eta)^T \widehat{\mathbf{W}} G(\theta, \eta)\right)^{-1} G(\theta, \eta)^T \widehat{\mathbf{W}} F \widehat{\mathbf{W}} G(\theta, \eta) \left(G(\theta, \eta)^T \widehat{\mathbf{W}} G(\theta, \eta)\right)^{-1},$$

*and where $G(\theta, \eta)$ is the Jacobian of the population moments at the ground truth parameter values $\theta, \eta$ and $F$ is the asymptotic variance of the sample moments.*

For optimal weight matrix in Equation 5, this simplifies to $V = (G(\theta, \eta)^T F^{-1} G(\theta, \eta))^{-1}$. These are standard results on GMM estimators, which follow by straightforwardly applying the results in Hansen [1982]. We remark that these asymptotic results require a set of conditions on the sample moments, which are slightly nuanced in this setting with multiple sources of information. We discuss these conditions and prove that they are satisfied in Appendix A for the setting of proxy and synthetic samples. Given this asymptotic behavior, we can derive valid confidence intervals for our parameter estimates.

### 4.6 Why does synthetic data improve performance?

To understand where the benefits arise from incorporating the proxy and synthetic data into our GMM estimator, we analyze the interaction between our moment conditions. Note that the functions $\psi$ are often referred to as "residuals" in the GMM literature; since $\psi(\theta)$ should be zero in-expectation, deviations from zero are interpretable as a kind of residual. The key intuition is that synthetic data will improve performance when the synthetic-data residuals are predictive of the real-data residuals.

First, we note that if the synthetic data were perfectly simulated, $X$ and $Y$ would be perfectly recovered from the unlabeled text $T$. With ground truth $X, Y$, we can perfectly recover the residual terms. In settings where we have good but imperfect simulations, $\hat{X}, \hat{Y}$ and $\tilde{X}, \tilde{Y}$ are highly correlated with the errors in the true data, and we can approximately estimate the real-data residuals with the synthetic data. Within our GMM-based approach, this is all handled implicitly in our two-step

estimation procedure. During the first estimation step, each set of parameters (e.g., defined on the observed, proxy, and synthetic data) is independently identified since the initial weighting is an identity matrix. The key insight is that, during the second estimation step, the weighting matrix $\widehat{\mathbf{W}}$, which is the inverse of the moment covariance matrix, captures the interactions between the observed residual terms and the residuals from the synthetic data in our GMM objective. This is captured in the off-diagonal terms of the moment covariance matrix. Partitioning the moments into observed data residuals $m_t(\theta)$ and synthetic data residuals $h_t(\eta)$, we derive an explicit formula for the asymptotic variance of $\sqrt{T}(\hat{\theta}_T - \theta)$ in Theorem 1 with the full proof in Appendix D.

**Theorem 1.** *The asymptotic variance of $\sqrt{T}(\hat{\theta}_T - \theta)$ is given by*

$$\left(\frac{d\mathbb{E}[m(\theta)]}{d\theta'} A \frac{d\mathbb{E}[m(\theta)]}{d\theta} - \left(\frac{d\mathbb{E}[m(\theta)]}{d\theta'} B \frac{d\mathbb{E}[h(\eta)]}{d\eta}\right) \left(\frac{d\mathbb{E}[h(\eta)]}{d\eta'} D \frac{d\mathbb{E}[h(\eta)]}{d\eta}\right)^{-1} \left(\frac{d\mathbb{E}[h(\eta)]}{d\eta'} B^\top \frac{d\mathbb{E}[m(\theta)]}{d\theta}\right)\right)^{-1}.$$

*with $A, B, D, m(\theta), h(\eta)$ defined in Appendix D.*

We find two important conclusions. First, when these residuals are independent of the observed data, the formula reduces to the optimal variance based only on the fully observed data. That is, in the worst case where synthetic data is completely uninformative, including it does not hurt (at least asymptotically). Second, when the real and synthetic residuals are correlated (as we would hope), we derive a lower bound on the variance which is proportional to the residual variance in a regression of the observed data residuals on the span of the synthetic data residuals. This bound is minimized by choosing moments that span the conditional expectation of the observed data residuals given $T_i$, a sufficient condition for which is that the conditional distribution of $\hat{X}, \hat{Y}$ or $\tilde{X}, \tilde{Y}$ given $T$ equals the conditional distribution of $X, Y$.

# 5 Experimental Results

## 5.1 Baselines

Existing methods in the literature are well-studied in the context of predicted outcomes and more recently, predicted covariates. However, it remains unclear how to aggregate information from fully synthetic data. We consider how to adapt existing debiasing methods; PPI++ [Angelopoulos et al., 2023b] and recent variants [Ji et al., 2025].

**RePPI** The most general approach is perhaps given by RePPI [Ji et al., 2025], which predicts the optimal loss through fitting an arbitrary model that maps the proxy and synthetic loss to the real loss. This results in the objective defined in Proposition 2. The resulting estimate retains asymptotic normality conditions (see Appendix E.2.1 for the proof and algorithm details).

**PPI++Proxy, PPI++Synth** To adapt PPI++ [Angelopoulos et al., 2023b] to this setting, we take an instantiation of RePPI, where the model is a convex combination to limit the number of parameters. This results in the following objective defined in Proposition 3. We refer to the estimator with $\alpha = 1$ as PPI++Proxy, as the synthetic terms vanish, yielding an estimator that combines real and proxy data. We refer to the estimator with tunable $\alpha \in [0, 1]$ as PPI++Synth, which combines real, proxy, and synthetic data. Note that the addition of this hyperparameter $\alpha$ adds increased complexity, and techniques such as cross-fitting must be used to select it in a statistically valid fashion. The resulting estimate retains asymptotic normality conditions (see Appendix E.2.2 for the proof and algorithm details).

**PPI++Synth (Oracle)** As an upper bound, we conduct a grid search over different possible $\alpha$ values *without* cross-fitting. Note, this is not a valid solution in the setup, as it requires peeking in hyperparameter selection, but it provides an oracle version of the baseline for reference.

## 5.2 Experimental Setup

**Datasets.** We validate the finite-sample performance of our estimator for logistic regression and ordinary least squares (OLS) regression on the following 4 computational social science tasks: First, we use online requests posted on Stack Exchange and Wikipedia [Danescu-Niculescu-Mizil et al.,

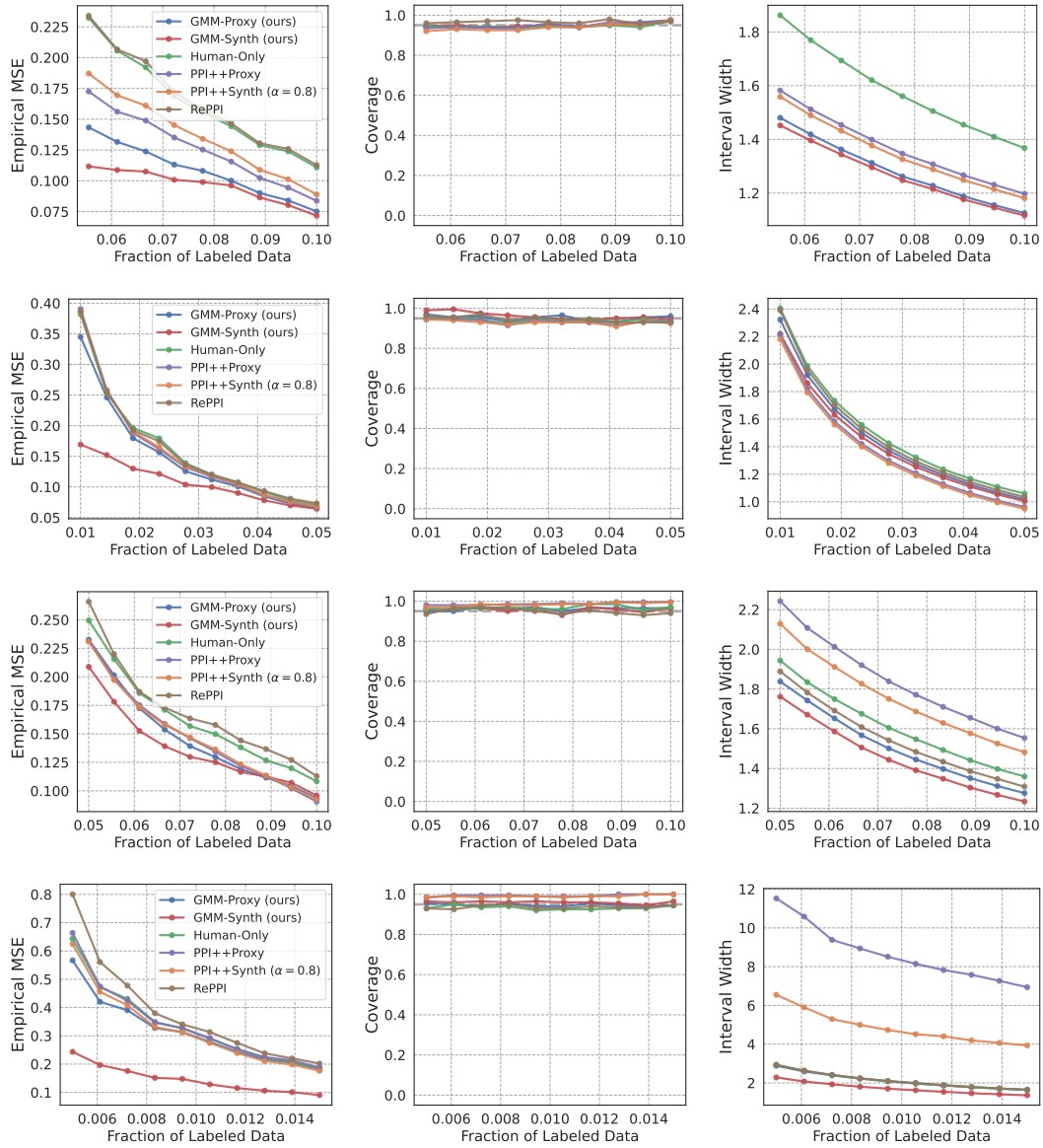

Figure 1: **Main Results (Logistic regression)**. We observe large reductions in MSE, especially in very low-label regimes. Each row corresponds to a task (i.e., 1pp, Hedging, Stance, Congressional Bills Data (from top to bottom)); each column corresponds to a metric (i.e., MSE, coverage, confidence interval width (from left to right)). Note that we report the PPI++Synth oracle number for PPI++Synth (see Figure 8 in Appendix F for PPI++Synth with cross-fitting results). When the best performing PPI++Synth is equivalent to PPI++Proxy (i.e., $\alpha = 1$), we report the second-best performing PPI++Synth method. See Figure 6 in Appendix F for full grid-search results over different $\alpha$ values. Results are averaged over 200 trials.

2013b] to estimate how the presence of hedging markers (i.e., expressions of uncertainty) affect perceived politeness. Second, we use the same dataset to estimate how the usage of first-person plural pronouns affect perceived politeness. Third, we use a corpus of climate-related news headlines [Hmielowski et al., 2014] to estimate the effect of affirming linguistic devices on media stance toward global warming (i.e., whether the news headline supports or rejects climate change). Lastly, we use congressional bills texts [Adler and Wilkerson, 2011] to estimate the effect of a legislator's DW-Nominate measure [Lewis et al., 2024] of ideology on the type of bill (whether the bill pertains to macroeconomy). In all tasks, the target parameter is the regression coefficient corresponding to the explanatory variable of interest.

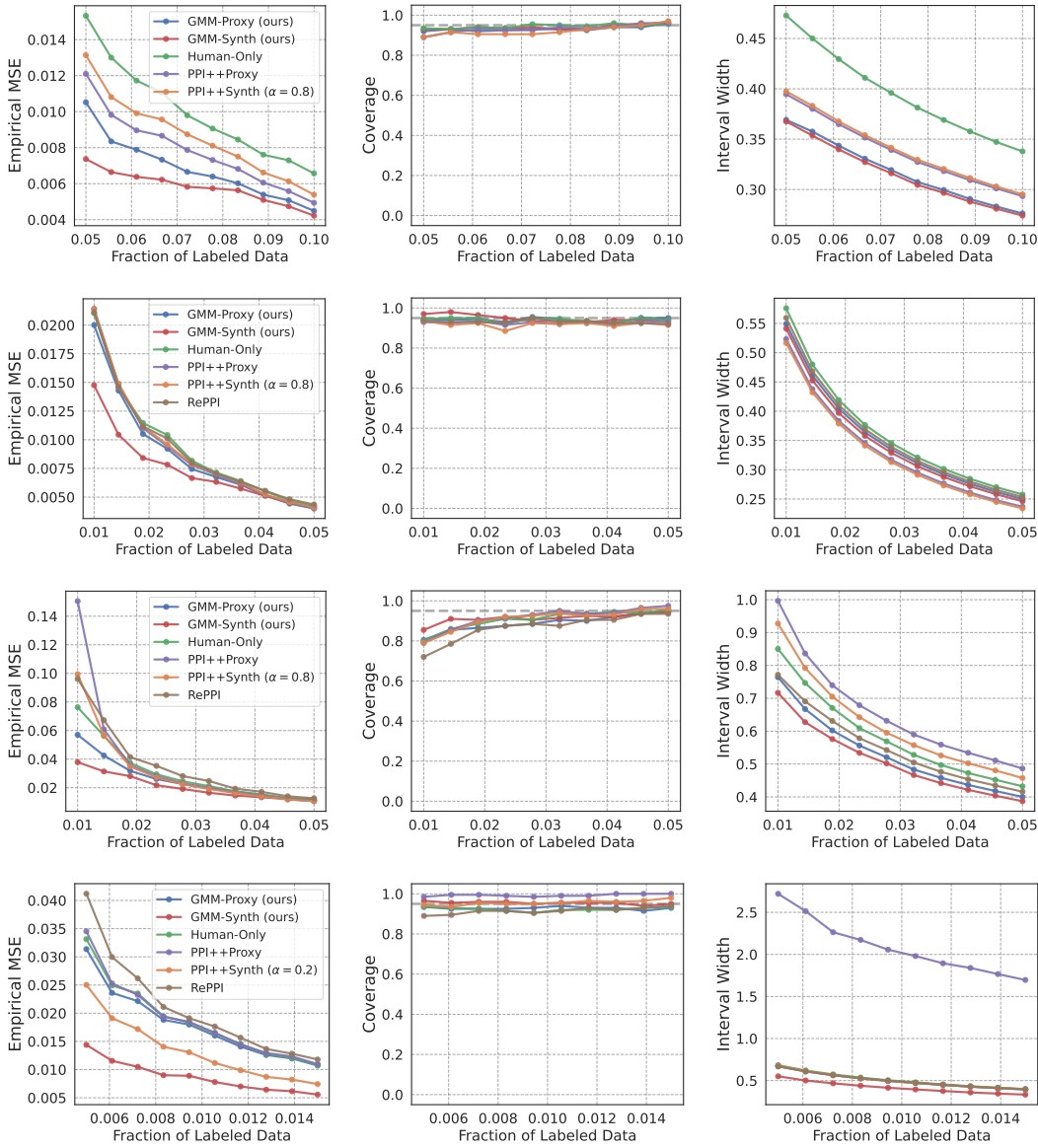

Figure 2: **Main Results (OLS)**. We again observe large reductions in MSE, especially in very low-label regimes. Each row corresponds to a task (i.e., 1pp, Hedging, Stance, Congressional Bills Data (from top to bottom)); each column corresponds to a metric (i.e., MSE, coverage, confidence interval width (from left to right)). Note that we report the PPI++Synth oracle number for PPI++Synth (see Figure 9 in Appendix F for PPI++Synth with cross-fitting results). When the best performing PPI++Synth is equivalent to PPI++Proxy (i.e., $\alpha = 1$), we report the second-best performing PPI++Synth method. See Figure 7 in Appendix F for full grid-search results over different $\alpha$ values. Results are averaged over 200 trials.

**Models and Metrics.** We use GPT-4o [Hurst et al., 2024] without any task-specific fine-tuning to generate both proxy and synthetic data. We also include additional results, using open-source, worse quality models (i.e., Llama-3-8b and Qwen-3-8b) in Appendix F (Figures 10, 11, 12, 13). We evaluate our method's performance against the adapted baselines discussed in Section 5.1 using four key metrics: empirical mean-squared error (MSE), coverage, confidence interval width, and effective sample size. The effective sample size represents the number of human-labeled samples that a classical estimator would require to achieve the same MSE as our method's estimate. In other words, this metric quantifies how many human annotations the method effectively saves while maintaining equivalent mean squared error.

### 5.3 Results

The results for our method's performance are shown in Figures 1 and 2. We will highlight some key observations. First, we observe that GMM-Synth achieves the lowest MSE, outperforming all baselines on 8 out of 8 downstream tasks. Notably, performance gains (of more than 50% reductions in MSE) are most pronounced when the fraction of labeled data is small, precisely the setting where the need for synthetic data is best motivated. Crucially, this does not come at a loss of validity of the parameter estimates; GMM-Synth retains valid coverage and results in tighter confidence intervals in 7 out of 8 downstream tasks. In Figures 4 and 5 (in Appendix F), we further observe that our method substantially improves effective sample size in data-limited settings. In other words, our method effectively saves large amounts of human annotations (up to more than 50%) while maintaining equivalent mean squared error. Second, GMM-Synth consistently exhibits gains over GMM-Proxy across all considered tasks. This demonstrates that synthetic data provides additional benefits beyond those of proxy-labeled data, and that our method effectively integrates these multiple sources, retaining their respective benefits. In other words, it shows how much additional benefit there is for the practitioner to not only use the model to label unlabeled samples but also use it to generate entirely new unlabeled samples to aid in statistical inference. Third, interestingly, unlike the additional gains demonstrated in GMM-Synth compared to GMM-Proxy, we observe that incorporating synthetic data via PPI++Synth (compared to PPI++Proxy) lead to benefits that are much less pronounced in 3 tasks, and, in fact, result in no gains in MSE in the remaining 5 tasks. This empirically demonstrates that our method is able to incorporate synthetic data much more effectively when compared to adaptations of existing debiasing methods in the literature. Importantly, we note that across all settings, using the proxy data and synthetic data alone yields greatly biased estimates (see Figure 3 in Appendix F). Further, we note that the same conclusions hold with even worse quality synthetic data generations from weaker language models; see Appendix F for results on open-source models such as LLaMA [Grattafiori et al., 2024] or Qwen [Yang et al., 2025].

## 6 Discussion

How synthetic data pipelines should be designed and implemented in practice hinges on reliable mechanisms for integrating information from them. In this work, we introduce a principled framework for reliably incorporating fully synthetic samples into downstream statistical analyses. We provide practical guidance for constructing synthetic samples from text-based foundation models in ways that support valid inference, and propose a new estimator based on generalized method of moments (GMM) estimation, where the key intuition is that synthetic data will improve performance when the synthetic-data residuals are predictive of the real-data residuals. Across the studied inferential tasks, we indeed observe a large degree of improvement in estimation, especially in very low-label regimes. More broadly, this work takes a first step toward understanding how imperfect synthetic data from foundation models can systematically be leveraged to support valid inference and to make reliable downstream conclusions. With the increased adoption and growing capabilities of foundation models, pipelines that incorporate their outputs will only become more complex. Our method provides an easily extensible estimation framework that can safely integrate the increasing variety and quality of synthetic data sources.

**Limitations and future directions.** A potential limitation of our framework is its reliance on the quality of the generative model (e.g., an LLM), as expected. As with other debiasing approaches, very poor-quality synthetic data would yield little-to-no benefits in statistical efficiency. Moreover, our theoretical guarantees, like those of debiasing methods, hold asymptotically and thus may fail to hold in extremely low-data regimes, potentially leading to undercoverage of the target parameter.

## Acknowledgments and Disclosure of Funding

We thank Michael Oberst and Gati Aher for helpful discussions in developing this work. We also thank Santiago Cortes-Gomez for thoughtful comments on the manuscript. This work is supported by the AI Research Institutes Program funded by the National Science Foundation under AI Institute for Societal Decision Making (AI-SDM), Award No. 2229881.

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

## A Conditions for Consistency and Asymptotic Normality

We provide a discussion about the necessary conditions for a GMM estimator to be consistent and asymptotically normal, showing that these conditions are indeed met for our augmented GMM.

As mentioned in the construction of our estimator, we define one moment condition for each parameter on the observed data $D$. We also define two moments for each parameter on the proxy and synthetic data. This leads to an overidentified system, with more moments than parameters, ensuring that the target parameter is identifiable.

Next, we establish a few conditions for valid asymptotic properties of our GMM estimator, specifically about the convergence and distributions of the stacked vector of the sample moments $g_t(\theta^*, \eta^*)$ at its optimum, which we will refer to as $g_t$ for brevity. First, we require that this vector of moments converges to its expectation, or that

$$\frac{1}{N} \sum_{t=1}^{N} g_t \rightarrow \mathbb{E}[g_t],$$

where $N = n + m$ is our total amount of data. Next, all moments must jointly respect the central limit theorem, or that

$$\sqrt{N} \left( \frac{1}{N} \sum_{t=1}^{N} g_t \right) \xrightarrow{d} \mathcal{N}(0, F),$$

where $F$ is some finite covariance matrix of all the moments $g_t$.

Under these standard regularity conditions on the moment vector $g_t$ [Newey and McFadden, 1994], these conditions are immediately satisfied for the moments defined on observed data, as each observation of the moments is independent. The same holds for the moments defined on proxy data, since $\hat{X}, \hat{Y}$ are functions of independent inputs $T$, and are therefore also independent across observations. The case of synthetic data is slightly more nuanced, but we show that the required conditions still hold, through the following lemma.

**Lemma 1.** *Let $\{\phi_j\}_{j=1}^{N}$ represent observations of the subset of moments corresponding to parameters of the synthetic data, and assume $\mathbb{E}\|\phi_j\|^2 < \infty$. Then, they are i.i.d., and consequently*

$$\frac{1}{N} \sum_{j=1}^{N} \phi_j \rightarrow \mathbb{E}[\phi_j] \quad and \quad \sqrt{N} \left( \frac{1}{N} \sum_{j=1}^{N} \phi_j \right) \xrightarrow{d} \mathcal{N}(0, \Sigma(\phi_j)),$$

*where $\Sigma(\phi_j)$ is the covariance matrix of $\phi_j$.*

*Proof.* We begin by noting that texts $\{T_j\}_{j=1}^{N}$ are drawn i.i.d. from the marginal distribution $\mathcal{D}_T$. For each $T_j$, a synthetic text $\tilde{T}_j$ is produced by a generative model (i.e., by an LLM), which uses independent randomness for each call. The model is conditioned only on an individual sample $(T_j, X_j)$ if $j$ is labeled or $(T_j, \hat{X}_j)$ otherwise. Since the generative process for each $T_j$ is independent and the mapping $\tilde{T}_j \mapsto (\tilde{X}_j, \tilde{Y}_j)$ is applied identically to each sample, the resulting pairs $(\tilde{X}_j, \tilde{Y}_j)$ are also i.i.d. As these pairs are drawn i.i.d., then the conditions are met via the central limit theorem. □

This result shows that the required conditions on the sample moments hold in our setting of proxy and synthetic samples; under the regularity conditions of Newey and McFadden [1994] Theorem 3.2, one immediately obtains Proposition 1 on the asymptotic behavior of our GMM estimator.

## B Asymptotic Efficiency

From Chamberlain [1987], we know that the lower bound on asymptotic variance among all regular estimators based on the moment restrictions is precisely the asymptotic variance that can be achieved in the general case by using the GMM estimator. Specifically, it achieves the semiparametric efficiency bound–the smallest possible asymptotic variance attainable by any regular estimator using these moment conditions. This corresponds to the local asymptotic minimax risk over all statistical models such that these moment equalities hold, in the sense that for any $\nu \in \mathbb{R}^d$

$$\lim_{N \to \infty} \inf_{\tilde{\theta} \text{ measurable}} \sup_{(\theta, \tilde{\mathcal{D}}) \in \Gamma(\theta^*, \mathcal{D})} \mathbb{E}_{\tilde{\mathcal{D}}}[[\nu^T \sqrt{N}(\tilde{\theta} - \theta)]^2] \geq \nu^T (G(\theta^*, \eta^*)^T F^{-1} G(\theta^*, \eta^*))^{-1} \nu$$

where $\Gamma(\theta^*, \mathcal{D})$ is any local neighborhood of true parameter $\theta^*$ and data-generating distribution $\mathcal{D}$ satisfying the moment conditions and regularity conditions $C_1$ in Chamberlain [1987] for parameter $\theta$.

*Proof.* Follows directly from Chamberlain [1987] Theorem 2. $\square$

In words, this indicates that no estimator can achieve lower asymptotic variance than GMM uniformly over all local distributions satisfying the same moment conditions.

## C   Moment Conditions

We provide a concrete example of our moment construction for the case of generalized linear models (GLMs) in two-dimensions.

### C.1   Example 1. Generalized Linear Models

Recall that the standard GLM formulation optimizes the objective function,

$$\ell_\theta(x, y) = -yx^T\theta + f(x^T\theta),$$

where $f$ is a function that is convex and smooth. We remark that this recovers the setting of logistic regression when $f(z) = \log(1 + \exp(z))$. Let us assume a two-dimensional setting for illustration. This translates to the population moment conditions of

$$\mathbb{E}\left[X_1 Y - \frac{\partial f(X^T\theta^*)}{\partial \theta_1}\right] = 0, \quad \mathbb{E}\left[X_2 Y - \frac{\partial f(X^T\theta^*)}{\partial \theta_2}\right] = 0$$

We have similar moments for proxy and synthetic data, where we use parameters $\eta = (\eta^{(1)}, \eta^{(2)})$, which are also two-dimensional. Within our GMM framework, we construct the following set of moment conditions across the observed, proxy, and synthetic data.

$$g_t(\theta, \eta) = \begin{bmatrix} s_t \\ s_t \\ s_t \\ s_t \\ s_t \\ s_t \\ s_t \\ 1 \\ 1 \\ 1 \\ 1 \end{bmatrix} \odot \begin{bmatrix} X_{t,1}Y_t - \frac{\partial f(X_t^T\theta)}{\partial \theta_1} \\ X_{t,2}Y_t - \frac{\partial f(X_t^T\theta)}{\partial \theta_2} \\ \hat{X}_{t,1}\hat{Y}_t - \frac{\partial f(\hat{X}_t^T\eta^{(1)})}{\partial \eta_1^{(1)}} \\ \hat{X}_{t,2}\hat{Y}_t - \frac{\partial f(\hat{X}_t^T\eta^{(1)})}{\partial \eta_2^{(1)}} \\ \tilde{X}_{t,1}\tilde{Y}_t - \frac{\partial f(\tilde{X}_t^T\eta^{(2)})}{\partial \eta_1^{(2)}} \\ \tilde{X}_{t,2}\tilde{Y}_t - \frac{\partial f(\tilde{X}_t^T\eta^{(2)})}{\partial \eta_2^{(2)}} \\ \hat{X}_{t,1}\hat{Y}_t - \frac{\partial f(\hat{X}_t^T\eta^{(1)})}{\partial \eta_1^{(1)}} \\ \hat{X}_{t,2}\hat{Y}_t - \frac{\partial f(\hat{X}_t^T\eta^{(1)})}{\partial \eta_2^{(1)}} \\ \tilde{X}_{t,1}\tilde{Y}_t - \frac{\partial f(\tilde{X}_t^T\eta^{(2)})}{\partial \eta_1^{(2)}} \\ \tilde{X}_{t,2}\tilde{Y}_t - \frac{\partial f(\tilde{X}_t^T\eta^{(2)})}{\partial \eta_2^{(2)}} \end{bmatrix}$$

## D   Partitioned GMM Asymptotic Variance

We now derive the asymptotic variance of our GMM estimator for specifically the target parameter $\hat{\theta}_T$.

**Theorem 1.** *The asymptotic variance of $\sqrt{T}(\hat{\theta}_T - \theta)$ is given by*

$$\left(\frac{d\mathbb{E}[m(\theta)]}{d\theta'} A \frac{d\mathbb{E}[m(\theta)]}{d\theta} - \left(\frac{d\mathbb{E}[m(\theta)]}{d\theta'} B \frac{d\mathbb{E}[h(\eta)]}{d\eta}\right) \left(\frac{d\mathbb{E}[h(\eta)]}{d\eta'} D \frac{d\mathbb{E}[h(\eta)]}{d\eta}\right)^{-1} \left(\frac{d\mathbb{E}[h(\eta)]}{d\eta'} B^\top \frac{d\mathbb{E}[m(\theta)]}{d\theta}\right)\right)^{-1}.$$

*with $A, B, D, m(\theta), h(\eta)$ defined in Appendix D.*

*Proof.* With the optimal choice of weight matrix for the full GMM estimation problem, the asymptotic variance of the vector $(\hat{\theta}, \hat{\eta})$ converges to $(G^T F^{-1} G)^{-1}$. To obtain the variance for $\hat{\theta}$ specifically, partition the moments into $g_t(\theta, \eta) = (m_t(\theta)', h_t(\eta)')'$, where $m_t(\theta) = S_t \odot \psi(\theta)$, and

$$h_t(\eta) = \begin{bmatrix} S_t \\ S_t \\ \vdots \\ S_t \\ 1 \\ \vdots \\ 1 \end{bmatrix} \odot \begin{bmatrix} \psi(\eta^{(1)}) \\ \vdots \\ \psi(\eta^{(M)}) \\ \psi(\eta^{(1)}) \\ \vdots \\ \psi(\eta^{(M)}) \end{bmatrix}$$

Given this partitioning, we can express

$$G(\theta, \eta) = \begin{bmatrix} \frac{d\mathbb{E}[m(\theta)]}{d\theta} & 0 \\ 0 & \frac{d\mathbb{E}[h(\eta)]}{d\eta} \end{bmatrix}$$

$$F = \begin{bmatrix} \mathbb{E}[m_t(\theta)m_t(\theta)'] & \mathbb{E}[m_t(\theta)h_t(\eta)'] \\ \mathbb{E}[h_t(\eta)m_t(\theta)'] & \mathbb{E}[h_t(\eta)h_t(\eta)'] \end{bmatrix}$$

By the partitioned inverse formula, we can express $F^{-1}$ as

$$\begin{bmatrix} A & B \\ B^\top & D \end{bmatrix}$$

where the upper left block $A$ is

$$\left(\mathbb{E}[m_t(\theta)m_t(\theta)'] - \mathbb{E}[m_t(\theta)h_t(\eta)']\mathbb{E}[h_t(\eta)h_t(\eta)']^{-1}\mathbb{E}[h_t(\eta)m_t(\theta)']\right)^{-1}$$

This term can be interpreted as the inverse of the asymptotic residual variance of a regression of $m_t(\theta)$ on the span of the vector $h_t(\eta)$.

The lower right block $D$ is, symmetrically, the inverse of the asymptotic residual variance of a regression of $h_t(\eta)$ on the span of the vector $m_t(\theta)$:

$$\left(\mathbb{E}[h_t(\eta)h_t(\eta)'] - \mathbb{E}[h_t(\eta)m_t(\theta)']\mathbb{E}[m_t(\theta)m_t(\theta)']^{-1}\mathbb{E}[m_t(\theta)h_t(\eta)']\right)^{-1}$$

Finally, the off-diagonal term multiplies $A$ by the coefficient in a regression of $m$ on $h$:

$$B = -A\mathbb{E}[m_t(\theta)h_t(\eta)']\mathbb{E}[h_t(\eta)h_t(\eta)']^{-1}$$

For the full variance,

$$G^\top F^{-1} G = \begin{bmatrix} \frac{d\mathbb{E}[m(\theta)]}{d\theta'} A \frac{d\mathbb{E}[m(\theta)]}{d\theta} & \frac{d\mathbb{E}[m(\theta)]}{d\theta'} B \frac{d\mathbb{E}[h(\eta)]}{d\eta} \\ \frac{d\mathbb{E}[h(\eta)]}{d\eta'} B^\top \frac{d\mathbb{E}[m(\theta)]}{d\theta} & \frac{d\mathbb{E}[h(\eta)]}{d\eta'} D \frac{d\mathbb{E}[h(\eta)]}{d\eta} \end{bmatrix}$$

Applying the partitioned inverse formula again, the upper left block of $(G^\top F^{-1} G)^{-1}$, which gives exactly the asymptotic variance of $\sqrt{T}(\hat{\theta}_T - \theta)$, is equal to

$$\left(\frac{d\mathbb{E}[m(\theta)]}{d\theta'} A \frac{d\mathbb{E}[m(\theta)]}{d\theta} - \left(\frac{d\mathbb{E}[m(\theta)]}{d\theta'} B \frac{d\mathbb{E}[h(\eta)]}{d\eta}\right) \left(\frac{d\mathbb{E}[h(\eta)]}{d\eta'} D \frac{d\mathbb{E}[h(\eta)]}{d\eta}\right)^{-1} \left(\frac{d\mathbb{E}[h(\eta)]}{d\eta'} B^\top \frac{d\mathbb{E}[m(\theta)]}{d\theta}\right)\right)^{-1}.$$

This can be interpreted similarly as the inverse of the asymptotic variance of the residual prediction error from a regression of $A^{-1/2}\frac{dm(\theta)}{d\theta}$ onto the span of a weighted linear combination of terms in $\frac{dh(\eta)}{d\eta}$. $\qquad\square$

This formula can be used to derive several properties of the procedure.

**Corollary 1.** *When the moments corresponding to real and simulated observations are uncorrelated, $Cov(m_t(\theta), h_t(\eta)) = 0$, the asymptotic variance of $\sqrt{T}(\hat{\theta}_T - \theta)$ equals the asymptotic variance of the GMM estimator only using the moments $m_t$ corresponding to real data.*

*Proof.* When $\mathbb{E}[m_t(\theta)h_t(\eta)'] = 0$, $A = (\mathbb{E}[m_t(\theta)m_t(\theta)'])^{-1}$ and $B = 0$, and so the upper left block of $(G^\top F^{-1} G)^{-1}$ equals $\left(\frac{d\mathbb{E}[m(\theta)]}{d\theta'}(\mathbb{E}[m_t(\theta)m_t(\theta)'])^{-1}\frac{d\mathbb{E}[m(\theta)]}{d\theta}\right)^{-1}$, which is the optimal variance corresponding to only using moments $m$ by the claim of Appendix B. $\qquad\square$

**Corollary 2.** *The asymptotic variance of $\sqrt{T}(\hat{\theta}_T - \theta)$ is lower bounded in the positive semi-definite order by $\left(\frac{d\mathbb{E}[m(\theta)]}{d\theta'} A \frac{d\mathbb{E}[m(\theta)]}{d\theta}\right)^{-1}$, which is minimized when $A$ is maximized.*

*Proof.* Let $M$, $N$ be positive semi-definite matrices such that $M - N \succeq 0$. Then $(M - N)^{-1} \succeq M^{-1}$. Apply this fact to $M = \frac{d\mathbb{E}[m(\theta)]}{d\theta'} A \frac{d\mathbb{E}[m(\theta)]}{d\theta}$ and $N = \frac{d\mathbb{E}[m(\theta)]}{d\theta'} B \frac{d\mathbb{E}[h(\eta)]}{d\eta}\left(\frac{d\mathbb{E}[h(\eta)]}{d\eta'} D \frac{d\mathbb{E}[h(\eta)]}{d\eta}\right)^{-1}\frac{d\mathbb{E}[h(\eta)]}{d\eta} B^\top \frac{d\mathbb{E}[m(\theta)]}{d\theta}$ which are both positive semi-definite because they are symmetric quadratic forms.

For any p.s.d. $A_1, A_2$, if $A_1 \succeq A_2$, $\left(\frac{d\mathbb{E}[m(\theta)]}{d\theta'} A_1 \frac{d\mathbb{E}[m(\theta)]}{d\theta}\right)^{-1} \preceq \left(\frac{d\mathbb{E}[m(\theta)]}{d\theta'} A_2 \frac{d\mathbb{E}[m(\theta)]}{d\theta}\right)^{-1}$ by reverse monotonicity of the inverse in p.s.d. order, so larger $A$ corresponds to smaller lower bound on the variance. $\qquad\square$

Among choices of moment functions $h_t(\eta)$ that depend solely on $T_t$, $A$ is maximized in the positive semi-definite order when the span of $h_t(\eta)$ contains $\mathbb{E}[m(\theta)|T_t]$. A sufficient but not necessary condition for this is that for some $j \in 1 \ldots M$, the conditional moments of the simulation are identical to those of the real data:

$$E[\psi(\eta_j)|T_i] = E[\psi(\theta)|T_i]$$

This calibration condition is satisfied when the conditional distribution of the simulated data given $T$ equals that of the real data, which is a natural simulation target, though not required for valid inference.

# E    Experimental Details

## E.1    Resources and Licensing Details

**Compute Details**    Each experiment is run on a A6000 GPU. We evaluate and average performance over 200 random seeds for all experiments.

**Asset Licenses**    The assets used in our work are subject to the following licenses: Stack Exchange and Wikipedia Data [Danescu-Niculescu-Mizil et al., 2013a]: CC BY-NC-SA 3.0; Climate news headlines data [Luo et al., 2020]: CC BY-NC-SA 3.0; Congressional bills data [Adler and Wilkerson, 2011]: MIT license; PPI++ Codebase [Angelopoulos et al., 2023b]: MIT license.

### E.2 Baseline Details

#### E.2.1 RePPI Implementation

In adapting RePPI [Ji et al., 2025] to our setting, we can model the imputed loss function in PPI with a ML-based approach. While in their paper, they choose a particular form of

$$g_\theta(X, \hat{Y}) = \frac{1}{1+r} \theta^T s^*(X, \hat{Y}),$$

where $s^*$ is the conditional score function. In our setting, we do not have access to $X$ on unlabeled instances, meaning that our model of the conditional score must take in inputs of $s^*(\hat{X}, \hat{Y}, \tilde{X}, \tilde{Y})$. In the case for GLMs and if we have access to unlabeled instances $X$, we know that the score is given by $\nabla \ell_\theta(X, Y) = X(f'(X^T \theta) - \mathbb{E}[Y|X, \hat{Y}])$, where $f$ is as defined in Section C.1. In that setting, we would only need to model $\mathbb{E}[Y|X, \hat{Y}]$. However, in our setting where we only have access to proxy and synthetic data, we need to directly model $\nabla \ell_\theta(X, Y)$ or try to learn $\mathbb{E}[\nabla \ell_\theta(X, Y)|\hat{X}, \hat{Y}, \tilde{X}, \tilde{Y}]$, as we do not observe $X$ to use in our predictions on unlabeled data.

We note that in our experiments, the sample splitting approach proposed in RePPI performs poorly due to cross-fitting; each split has a size of $\frac{1}{3}$ of the number of labeled data to estimate (i) the ground truth parameter on one fold, (ii) learn the ML model on the second fold, and (iii) have an accurate target parameter estimate on the final fold. As such, to learn the ML model for the imputed loss in RePPI, we choose to adopt a linear regression model (defined over a small number of covariates), which satisfies the required Donsker conditions to enable us to avoid any requirements on sample splitting as in standard DML approaches [Van Der Vaart and Wellner, Chernozhukov et al., 2018].

Therefore, our implementation of RePPI is as follows: we (1) fit $\hat{\theta}$ by optimizing the human-only loss, (2) we optimize an linear regression model that learns to map $h : (\hat{X}, \hat{Y}, \tilde{X}, \tilde{Y}) \to \nabla \ell_{\hat{\theta}}(X, Y)$, and (3) we perform power tuning and produce our parameter estimate by minimizing the imputed loss that incorporates $\hat{\theta}$ and $h$, all on the full available data. We use the same linear regression in estimating the imputed loss; the exception to this is on the Congressional Bills dataset, where we use XGBoost as linear regression performs very poorly in estimating the score function.

**Proposition 2.** *The RePPI objective with multiple predicted covariates and outcomes is given by*

$$L^{RePPI}(\theta) := \frac{1}{n} \sum_{i=1}^n \ell_\theta(X_i, Y_i) - \left( \frac{1}{n} \sum_{i=1}^n g_\theta(\hat{X}_i, \hat{Y}_i, \tilde{X}_i, \tilde{Y}_i) - \frac{1}{N} \sum_{i=1}^N g_\theta(\hat{X}_i, \hat{Y}_i, \tilde{X}_i, \tilde{Y}_i) \right). \quad (6)$$

*where*

$$g_\theta(\hat{X}_i, \hat{Y}_i, \tilde{X}_i, \tilde{Y}_i) = \frac{1}{1+r} \theta^T s^*(\hat{X}_i, \hat{Y}_i, \tilde{X}_i, \tilde{Y}_i), \; s^*(\hat{X}_i, \hat{Y}_i, \tilde{X}_i, \tilde{Y}_i) = \mathbb{E}[\nabla \ell_{\theta^*}(X, Y)|\hat{X}_i, \hat{Y}_i, \tilde{X}_i, \tilde{Y}_i],$$

*and $\theta^*$ is the target parameter estimate. The resulting estimate retains asymptotic normality conditions.*

#### E.2.2 PPI++Proxy and PPI++Synth Implementation

We now present a discussion on our adapted debiasing-based approach from Proposition 3.

**Proposition 3.** *The adapted PPI++ objective with multiple predicted covariates and outcomes is given by*

$$L^{PP}(\theta) := \frac{1}{N} \sum_{i=1}^N [(1 - \alpha) \cdot \ell_\theta(\tilde{X}_i, \tilde{Y}_i) + \alpha \cdot \ell_\theta(\hat{X}_i, \hat{Y}_i)] \quad (7)$$

$$+ \frac{1}{n} \sum_{i=1}^n (\ell_\theta(X_i, Y_i) - [(1 - \alpha) \cdot \ell_\theta(\tilde{X}_i, \tilde{Y}_i) + \alpha \cdot \ell_\theta(\hat{X}_i, \hat{Y}_i)]). \quad (8)$$

*where the estimate retains asymptotic normality conditions (see Appendix E.2.2 for the proof and algorithm details).*

---

**Algorithm 1** Cross-Fitting for PPI⁺⁺Synth

---

**Require:**
 1: Labeled data $\mathcal{D} = \{(T_i, X_i, Y_i)\}_{i=1}^{n}$,
 2: Proxy data $\widehat{\mathcal{D}} = \{(T_j, \widehat{X}_j, \widehat{Y}_j)\}_{j=1}^{n+m}$,
 3: Synthetic data $\widetilde{\mathcal{D}} = \{(\widetilde{T}_j, \widetilde{X}_j, \widetilde{Y}_j)\}_{j=1}^{n+m}$,
 4: K folds
**Ensure:** Debiased estimate $\hat{\theta}_{\text{CF}}$
 5: Split $\mathcal{D}$ into folds $\{\mathcal{I}_1, \ldots, \mathcal{I}_K\}$
 6:
 7: **for** $k = 1, \ldots, K$ **do**
 8:     define train-fold $\mathcal{I}_{\text{train}} = \bigcup_{r \neq k} \mathcal{I}_r$
 9:     $\hat{\theta}_1^{-k} \leftarrow \arg\min_\theta L_{\text{PP}}^{-k}(\theta; 0)$              ▷ (1) initial fit on train-fold
10:
11:     $\hat{\alpha}^{-k} \leftarrow \arg\min_{\alpha \in [0,1]} L_{\text{PP}}^{-k}(\hat{\theta}_1^{-k}; \alpha)$      ▷ (2) select mixture weight $\alpha$ on train-fold)
12:
13:     $\hat{\theta}^k \leftarrow \arg\min_\theta L_{\text{PP}}^{k}(\theta; \hat{\alpha}^{-k})$        ▷ (3) final fit on held-out fold with chosen $\alpha$)
14:
15: **end for**
16: **return** $\hat{\theta}_{\text{CF}} = \dfrac{1}{K} \sum_{k=1}^{K} \hat{\theta}^k$

---

**Asymptotic Normality** First, it is relatively straightforward to show that this is an unbiased estimate of the true objective.

$$
\begin{aligned}
\mathbb{E}[L^{PP}(\theta)] &= (1 - \alpha) \cdot \mathbb{E}[\ell_\theta(\tilde{X}, \tilde{Y})] + \alpha \cdot \mathbb{E}[\ell_\theta(\hat{X}, \hat{Y})] \\
&\quad + \mathbb{E}[\ell_\theta(X, Y)] - \mathbb{E}[(1 - \alpha) \cdot \ell_\theta(\tilde{X}, \tilde{Y})] - \alpha \cdot \mathbb{E}[\ell_\theta(\hat{X}, \hat{Y})])] \\
&= \mathbb{E}[\ell_\theta(X, Y)].
\end{aligned}
$$

Note that this holds for any choice of the hyperparameter $\alpha$.

Under the same assumptions as in the PPI++ paper [Angelopoulos et al., 2023b] (e.g., that $\frac{n}{n+m} \to c$ for some constant $c$ and, in the case of generalized linear models, the Hessian is non-singular, we perform their same approach to power tuning), we recover the asymptotic normality guarantees of the parameter estimate (as in Corollary 1 from Angelopoulos et al. [2023b]).

**Hyperparameter Selection via Cross-fitting** The added complexity from these modified debiasing-based approaches arises from the hyperparameter $\alpha$. We now discuss an approach for selecting $\alpha$ by performing cross-fitting. As previously mentioned, we can treat $\alpha$ as a simple version of RePPI [Ji et al., 2025] where we fit a convex combination of proxy and synthetic losses.

Namely, we partition our available data into two splits. We select $\alpha$ on one fold by minimizing:

$$
\arg\min_{\alpha \in [0,1]} L^{PP}(\theta_1),
$$

where $\theta_1$ is defined as the solution to the naive minimzation of $\mathbb{E}[\ell_\theta(X, Y)]$ on the same split. This essentially captures picking the $\alpha$ that best combines the proxy and synthetic losses to best mimic the behavior of the standard loss function. We then take this optimal $\alpha$ and use it to produce a parameter estimate on the held-out fold. We aggregate these estimates as is standard in cross-fitting approaches. This process is outlined in Algorithm 1.

## E.3 Prompt Texts

We present the full text prompts that were used to generate proxy covariates and labels (for the proxy data) and synthetic data. Note that the prompts used to extract covariates and labels from the synthetic text are identical to those used for the proxy data.

**Politeness (First Plural Pronouns) - Covariates:**
Does the following text contain first person plural pronouns (e.g., we, us, our, ourselves)?
Output either yes or no.
Text: """
{content}
"""
**Answer:**

**Politeness (First Plural Pronouns) - Labels:**
Is the following text polite? Output either A or B. Output a letter only.
A) Polite
B) Impolite
Text: """
{content}
"""
**Answer:**

**Politeness (Hedging) - Covariates:**
Does the following text contain hedging devices—expressions that indicate uncertainty,
caution, or a lack of full commitment to a claim (e.g., may, might, could, would, possibly,
probably, perhaps, apparently, suggest, indicate, seem, appear, it is likely that, it seems that)?
Respond with yes or no only.
Text: """
{content}
"""
**Answer:**

**Politeness (Hedging) - Labels:**
Is the following text polite? Output either A or B. Output a letter only.
A) Polite
B) Impolite
Text: """
{content}
"""
**Answer:**

**Stance Dataset - Covariates:**
Does the following text contain any affirmative device words? Output either yes or no.
Text: """
{content}
"""
**Answer:**

**Stance Dataset - Labels:**
A statement can agree, be neutral, or disagree with the statement: "Climate change/global
warming is a serious concern". Classify the following statement into one of the three
categories. Output either A, B, or C. Output a letter only.
A) Agree
B) Neutral
C) Disagree
Statement: """
{content}
"""
**Answer:**

**Congressional Bills Dataset - Covariates:**
You are a political scientist familiar with the U.S. Congress and the DW-NOMINATE scoring system, which places legislators and legislation on a left-right ideological spectrum ranging approximately from -1 (most liberal) to +1 (most conservative). Below is the text of a proposed bill. Based on the policy content, language, and framing of the bill, estimate the DW-NOMINATE score that best represents its ideological position. Output a single nonzero float between -1 and +1 representing the estimated DW-NOMINATE score of the bill.
Bill: """
{content}
"""
**Answer:**

**Congressional Bills Dataset - Labels:**
Does the following text relate to the economy? Output either true or false.
Text: """
{content}
"""
**Label:**

---

**Synthetic Data Generation Prompts**

**Politeness (First Plural Pronouns)**
Consider texts taken from user requests on Stack Exchange or Wikipedia. Each text is labeled as either polite or impolite, and either contains or does not contain first-person plural pronouns. Below is an example that {x}:
**Example:** """
{example}
"""
Now, generate a new example of a request that also {x}.

**Politeness (Hedging)**
Consider texts taken from user requests on Stack Exchange or Wikipedia. Each text can be labeled as either polite or impolite, and as either containing a hedging device or not containing one. Hedging devices are expressions that indicate uncertainty, caution, or a lack of full commitment to a claim (e.g., may, might, could, would, possibly, probably, perhaps, apparently, suggest, indicate, etc.). Below is an example that {x}:
**Example:** """
{example}
"""
Now, generate a new example of a request that also {x}.

**Stance**
Consider news headlines that take a stance — agree, disagree, or neutral — on the statement: "Climate change/global warming is a serious concern."
Each headline also either contains or does not contain an affirmative device.
Below is an example of a headline.
**Example:** """
{example}
"""
Affirmative device: {x}
Now, generate a new news headline about global warming that also {x}.

## F   Additional Results

We present additional experimental results consisting of:

- Prediction accuracy of GPT-4o, Llama-3-8b, Qwen-3-8b (Think) for the covariates and outcomes of interest (Table 1)
- Performance of a naive estimator that *only* uses synthetic data (Figure 3)
- Effective sample size results (Figures 4 and 5)
- Grid search results for PPI++Synth (Oracle) across different $\alpha$ values (Figures 6 and 7)
- Cross-fitting results for PPI++Synth (Figures 8 and 9)
- Llama-3-8b results for logistic regression (Figure 10) and OLS (Figure 11)
- Qwen-3-8b results for logistic regression (Figure 12) and OLS (Figure 13)
- Results with zero-shot prompting for synthetic data generation (Figure 14)
- Results with using random noise for synthetic data (Figure 15)

Table 1: Accuracy of LLMs for prediction tasks. This represents the quality of the proxy covariates and proxy labels $(\hat{X}, \hat{Y})$. Note, for the Congressional Bills dataset, we report MSE for the X Prediction task since DW-NOMINATE scores are continuous values.

| Model | Hedging | | 1pp | | Stance | | Congressional Bills | |
|---|---|---|---|---|---|---|---|---|
| | X Pred. | Y Pred. | X Pred. | Y Pred. | X Pred. | Y Pred. | X Pred. | Y Pred. |
| GPT-4o | 0.764 | 0.785 | 0.994 | 0.785 | 0.864 | 0.743 | 0.184 | 0.827 |
| Llama-3-8b | 0.485 | 0.335 | 0.878 | 0.501 | 0.619 | 0.637 | 0.688 | 0.564 |
| Qwen-3-8b (Think) | 0.723 | 0.325 | 0.961 | 0.325 | 0.904 | 0.684 | 0.253 | 0.757 |

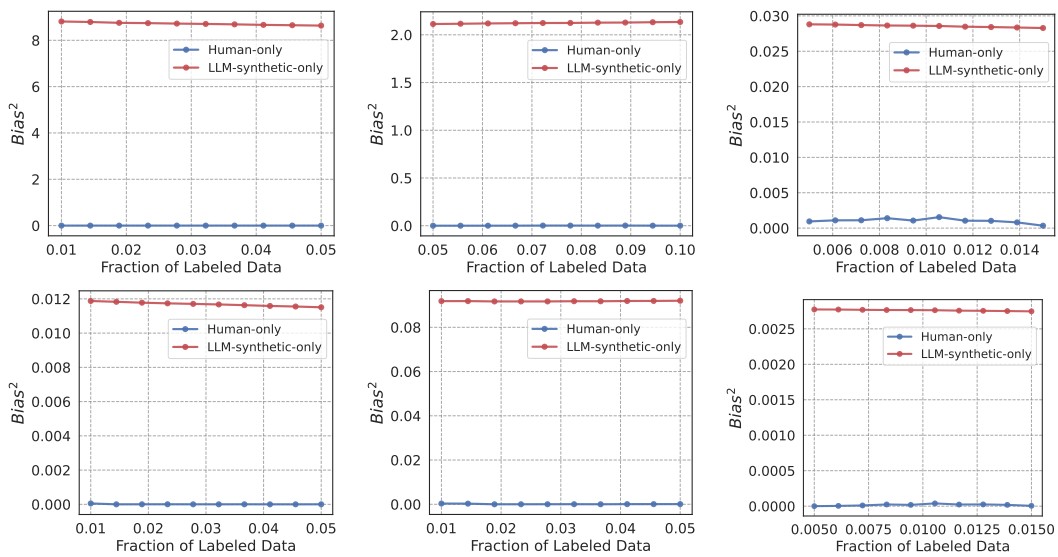

Figure 3: Performance of a naive estimator for logistic regression (top) and OLS (bottom) using synthetic data only (Politeness (Hedging), Stance, Congressional Bills (from left to right)). We clearly observe that naively using only synthetic data for the estimation task leads to largely biased estimates, as expected.

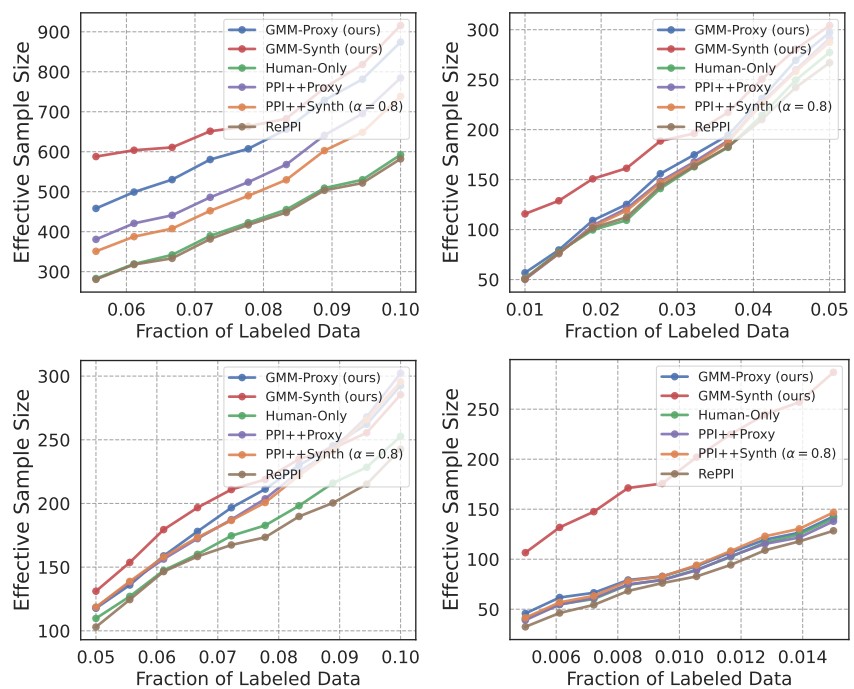

Figure 4: Effective sample size for logistic regression (Politeness (1pp), Politeness (Hedging), Stance, Congressional Bills (from left to right)). We observe large gains in effective sample size, up to more than 50%. This represents how many human annotations the method effectively saves while maintaining the same performance (in terms of mean squared error).

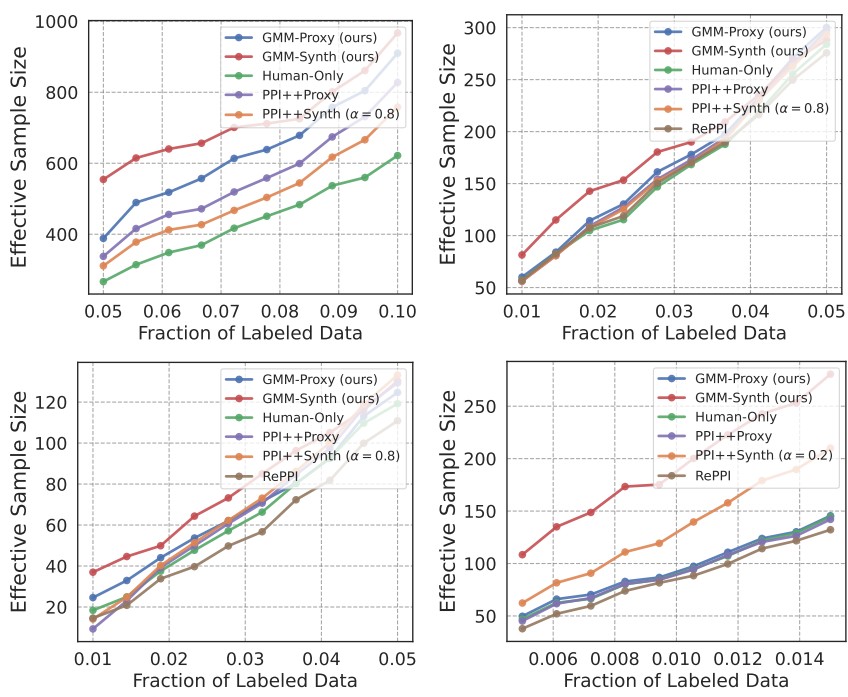

Figure 5: Effective sample size for OLS (Politeness (1pp), Politeness (Hedging), Stance, Congressional Bills (from left to right)). The RePPI method is omitted from the 1pp plot because its effective sample size drops too low.

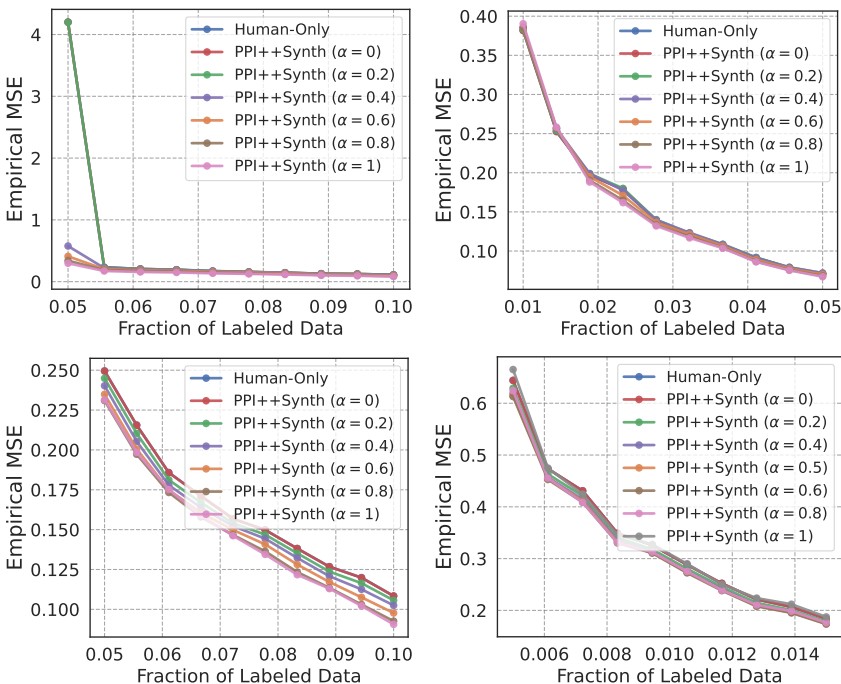

Figure 6: Grid search results for logistic regression (Politeness (1pp), Politeness (Hedging), Stance, Congressional Bills (from left to right)). This plot shows the grid search over different possible $\alpha$ values *without* cross-fitting. Note, this is not a valid solution in our setup, as it requires peeking in hyperparameter selection, but it provides an oracle version of the baseline, which we term as PPI++Synth (Oracle). The $\alpha$ value that leads to the smallest MSE is the one reported in Figure 1 in the main text.

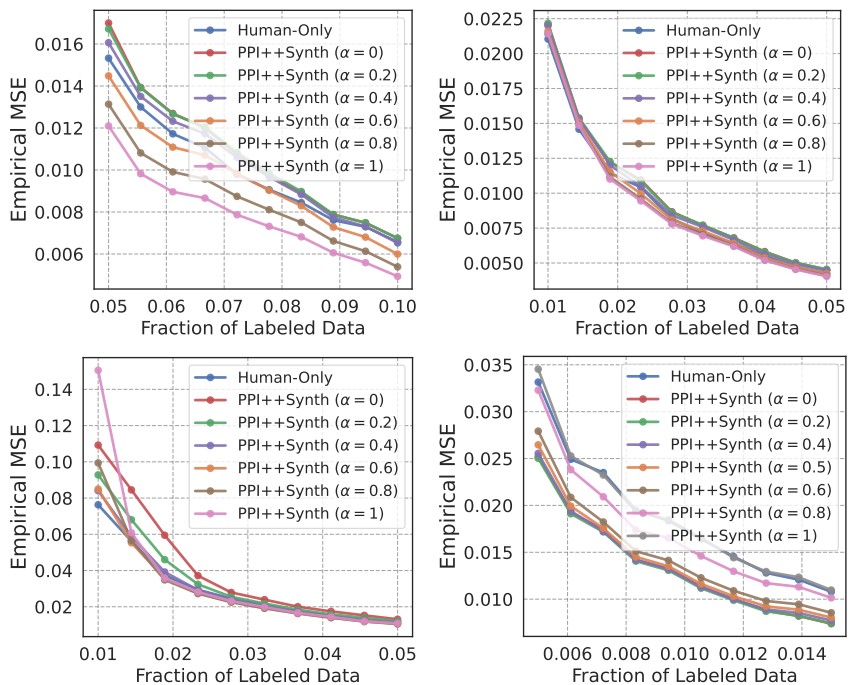

Figure 7: Grid search results for OLS (Politeness (1pp), Politeness (Hedging), Stance, Congressional Bills (from left to right)). The $\alpha$ value that leads to the smallest MSE is the one reported in Figure 2 in the main text.

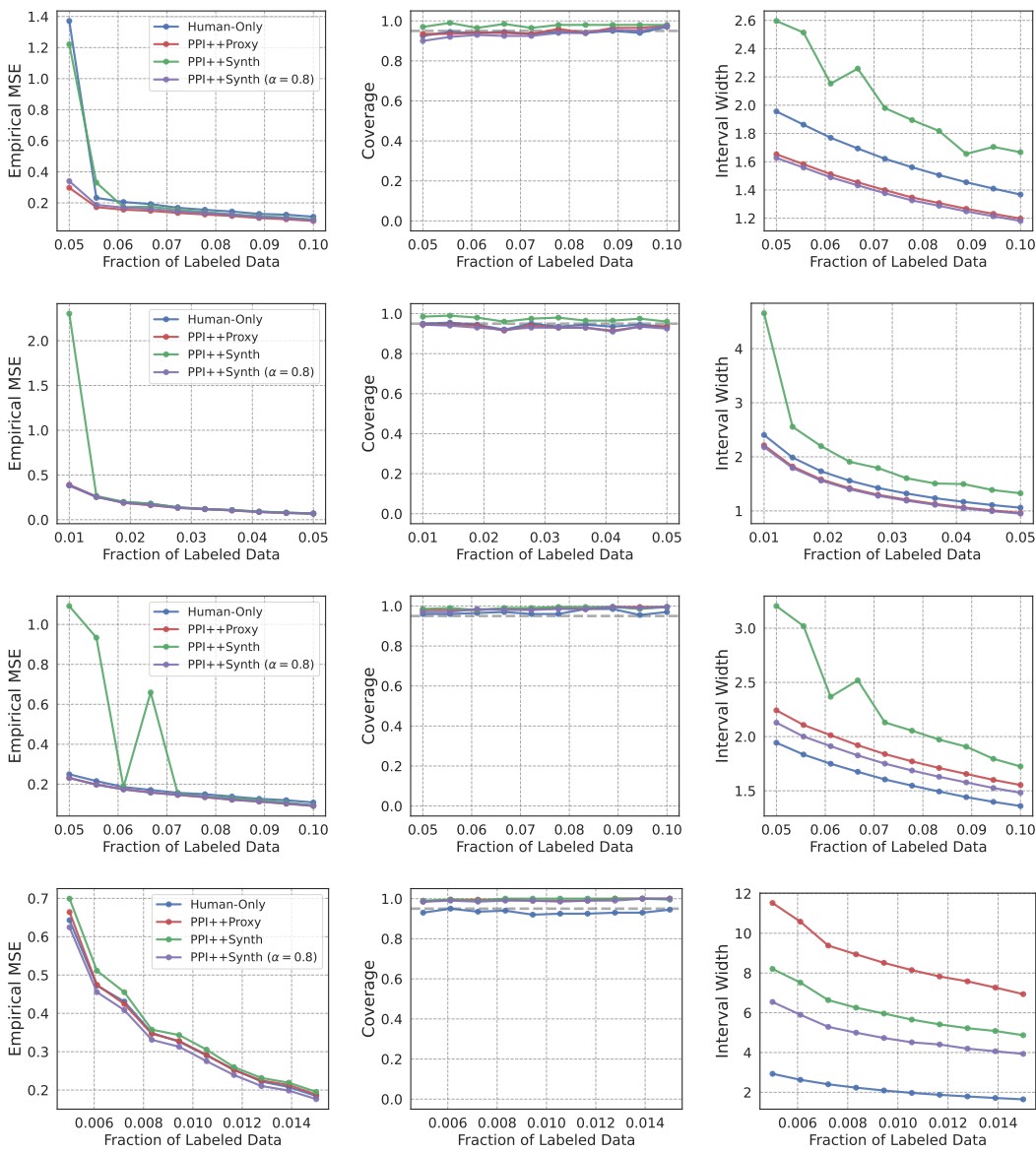

Figure 8: PPI++Synth results for logistic regression. This is the valid implementation with cross-fitting to select hyperparameters in a statistically valid fashion. We observe that it is upper-bounded by its oracle variant (PPI++Synth $\alpha = 0.8$), as expected. In the main text (Fig. 1), we report the oracle variant results to account for potential gains from improved cross-fitting techniques.

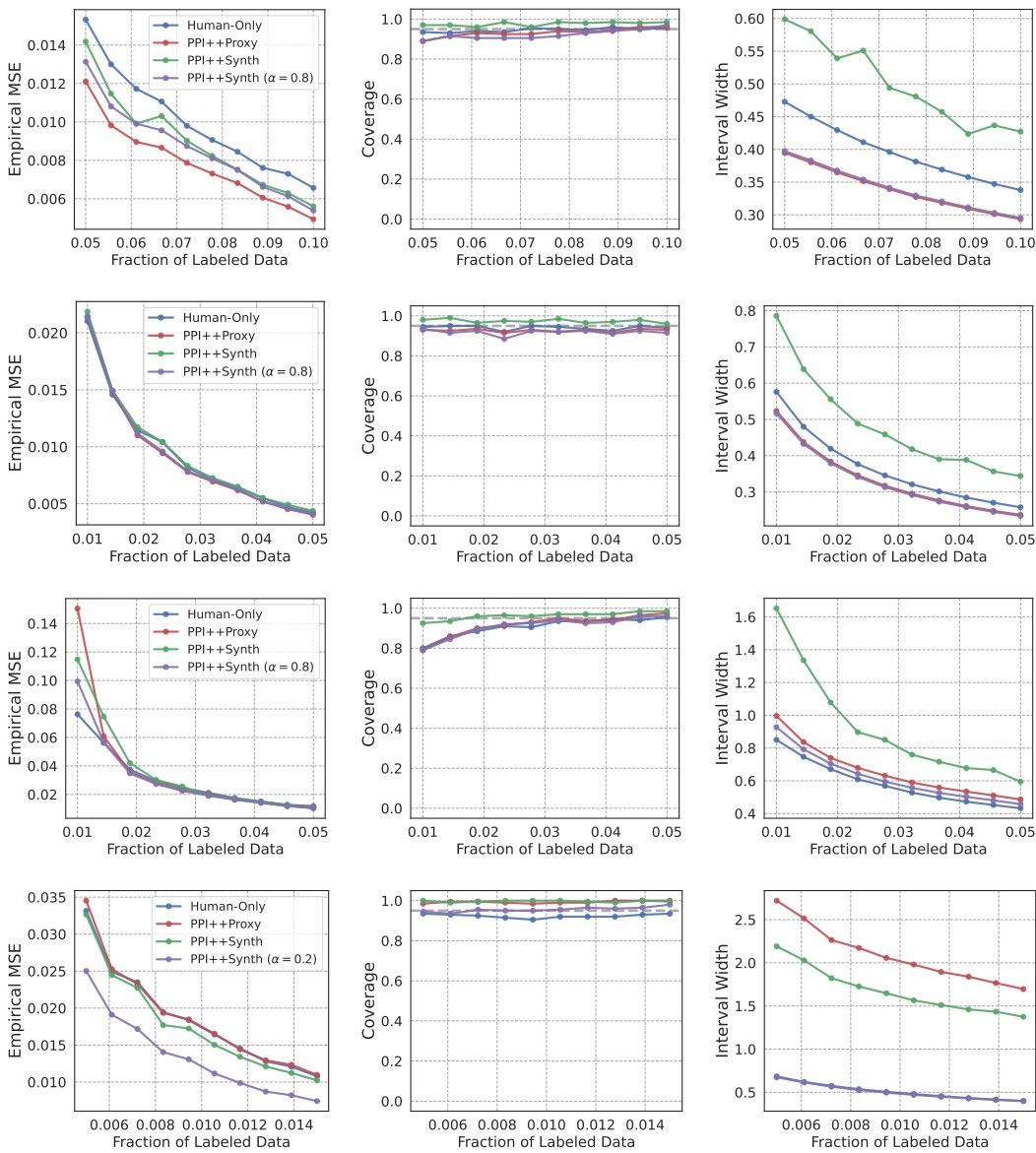

Figure 9: PPI++Synth results for OLS. Similarly as above, we observe that it is upper-bounded by its oracle variant (PPI++Synth $\alpha = 0.8$ and PPI++Synth $\alpha = 0.2$ for CBP), as expected.

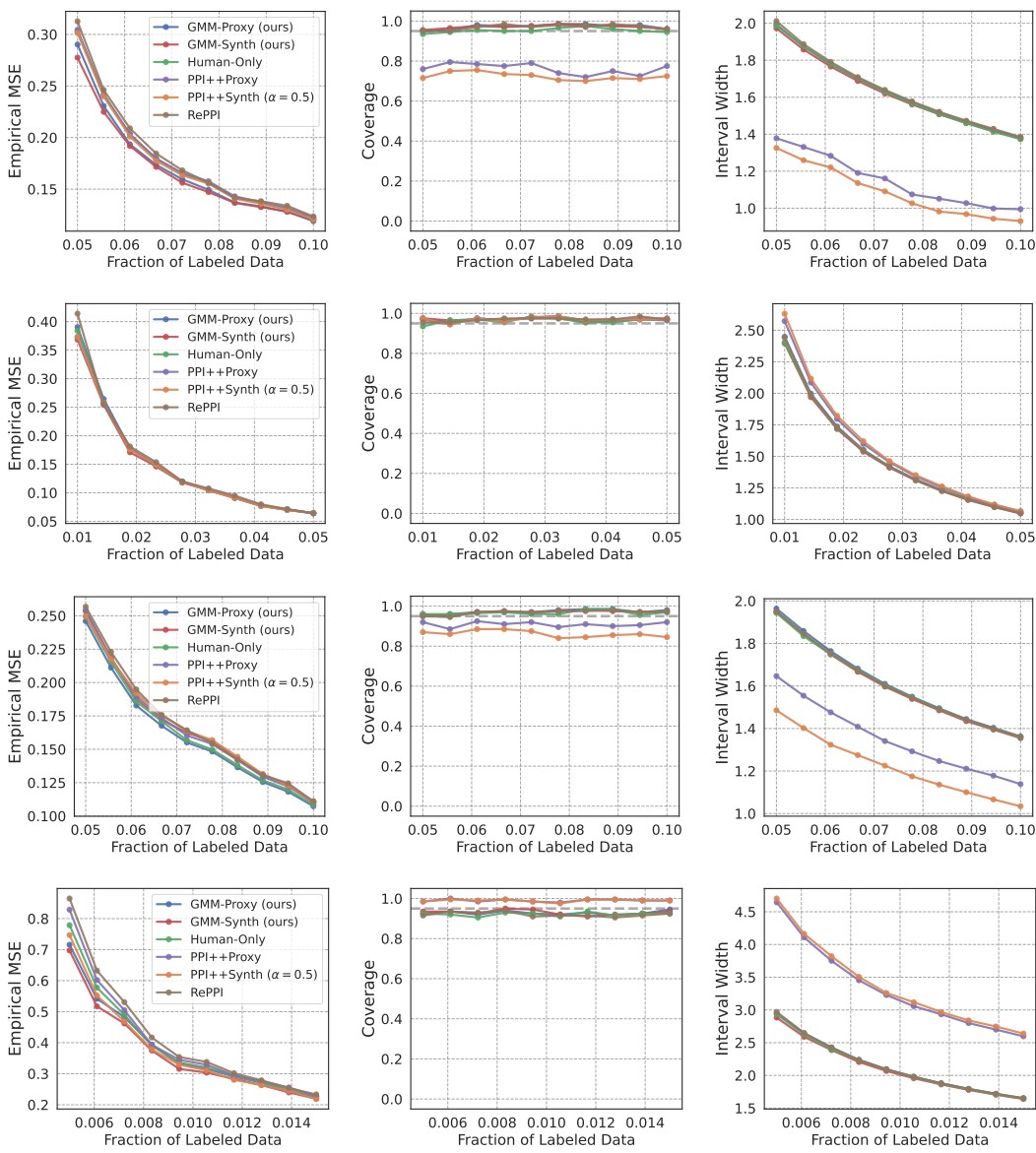

Figure 10: Llama-3-8b results for logistic regression. Each row corresponds to a task (i.e., 1pp, Hedging, Stance, Congressional Bills Data (from top to bottom)); each column corresponds to a metric (i.e., MSE, coverage, confidence interval width (from left to right)). Results are averaged over 200 trials.

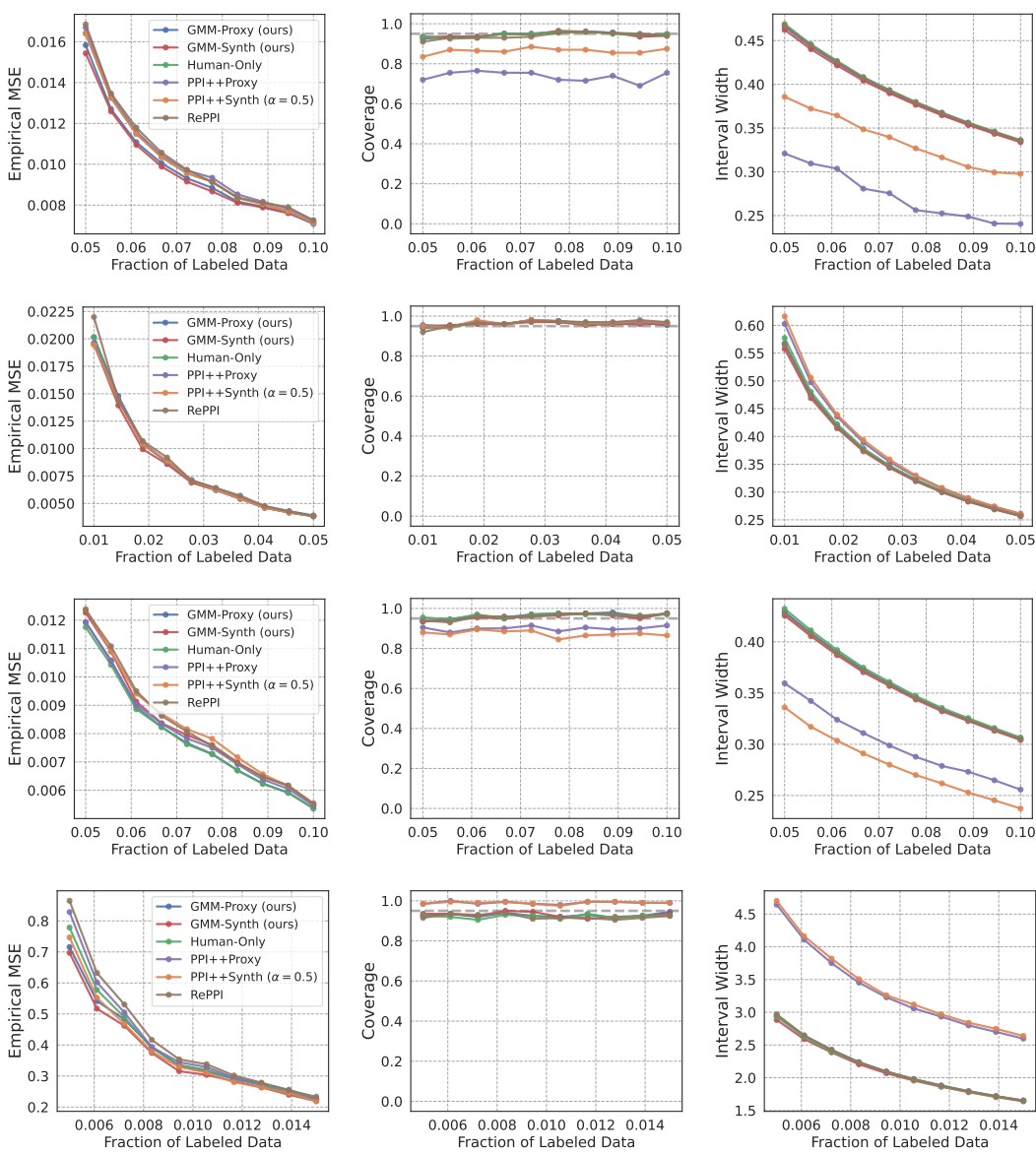

Figure 11: Llama-3-8b results for OLS. Each row corresponds to a task (i.e., 1pp, Hedging, Stance, Congressional Bills Data (from top to bottom)); each column corresponds to a metric (i.e., MSE, coverage, confidence interval width (from left to right)). Results are averaged over 200 trials.

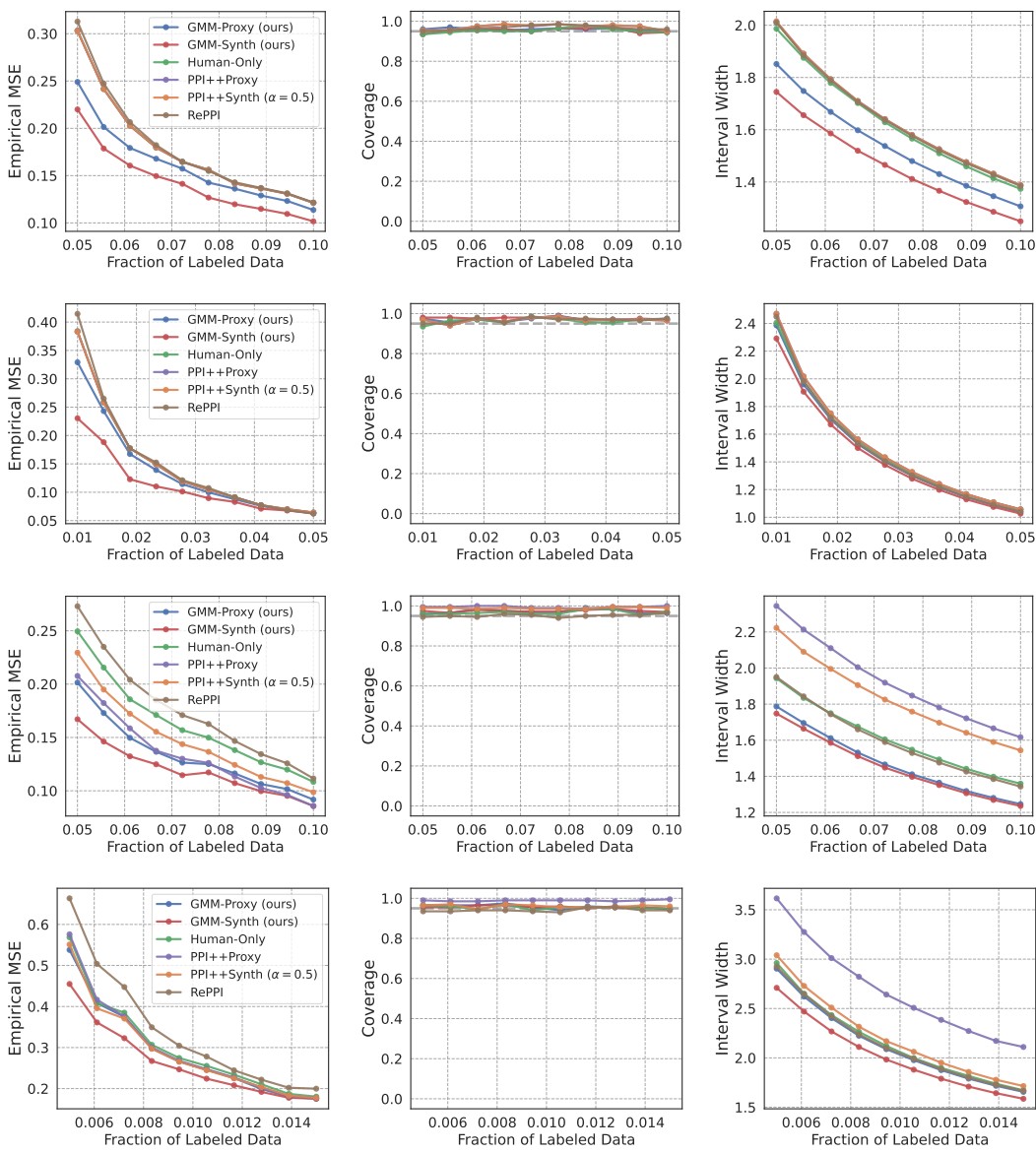

Figure 12: Qwen-3-8b results for logistic regression. Each row corresponds to a task (i.e., 1pp, Hedging, Stance, Congressional Bills Data (from top to bottom)); each column corresponds to a metric (i.e., MSE, coverage, confidence interval width (from left to right)). Results are averaged over 200 trials.

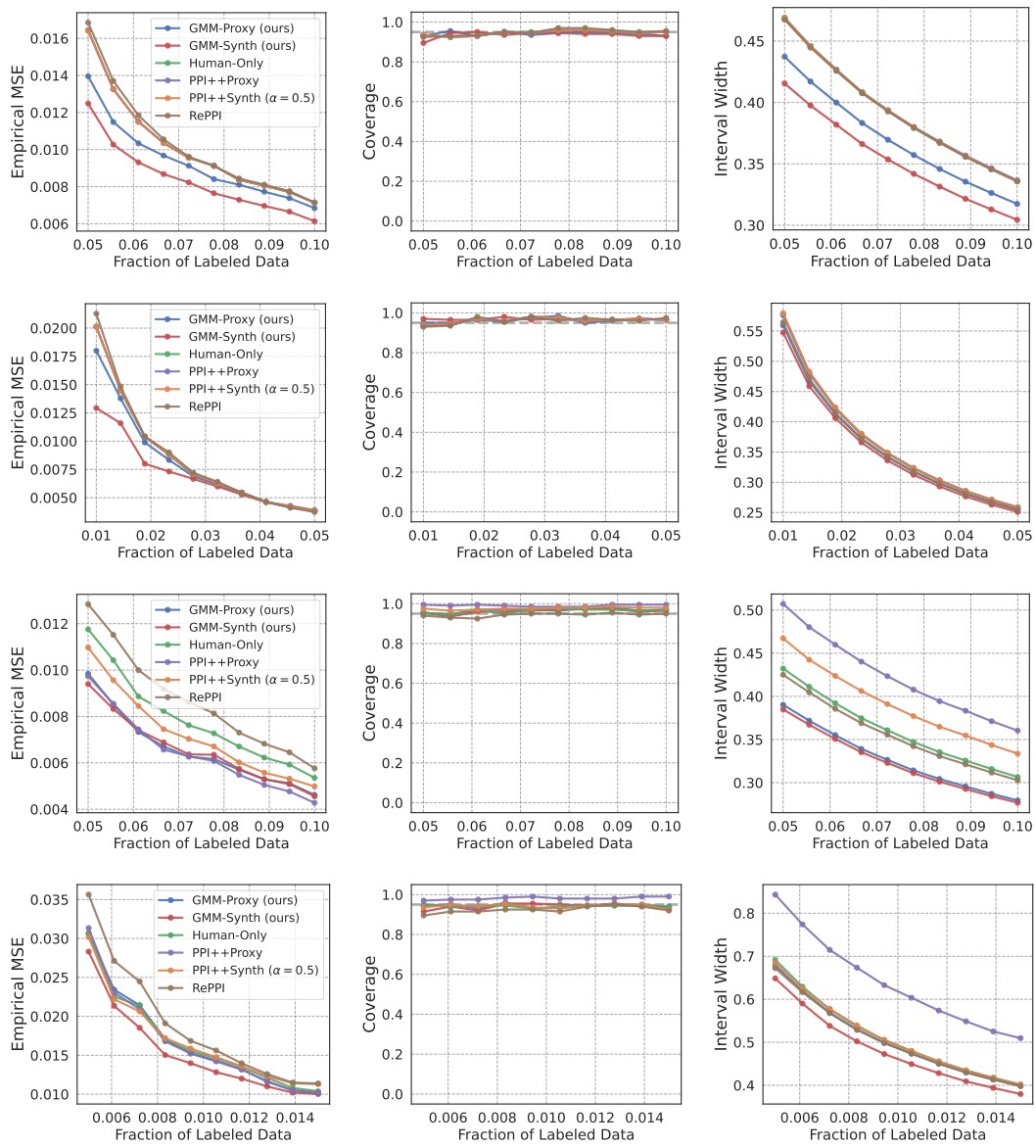

Figure 13: Qwen-3-8b results for OLS. Each row corresponds to a task (i.e., 1pp, Hedging, Stance, Congressional Bills Data (from top to bottom)); each column corresponds to a metric (i.e., MSE, coverage, confidence interval width (from left to right)). Results are averaged over 200 trials.

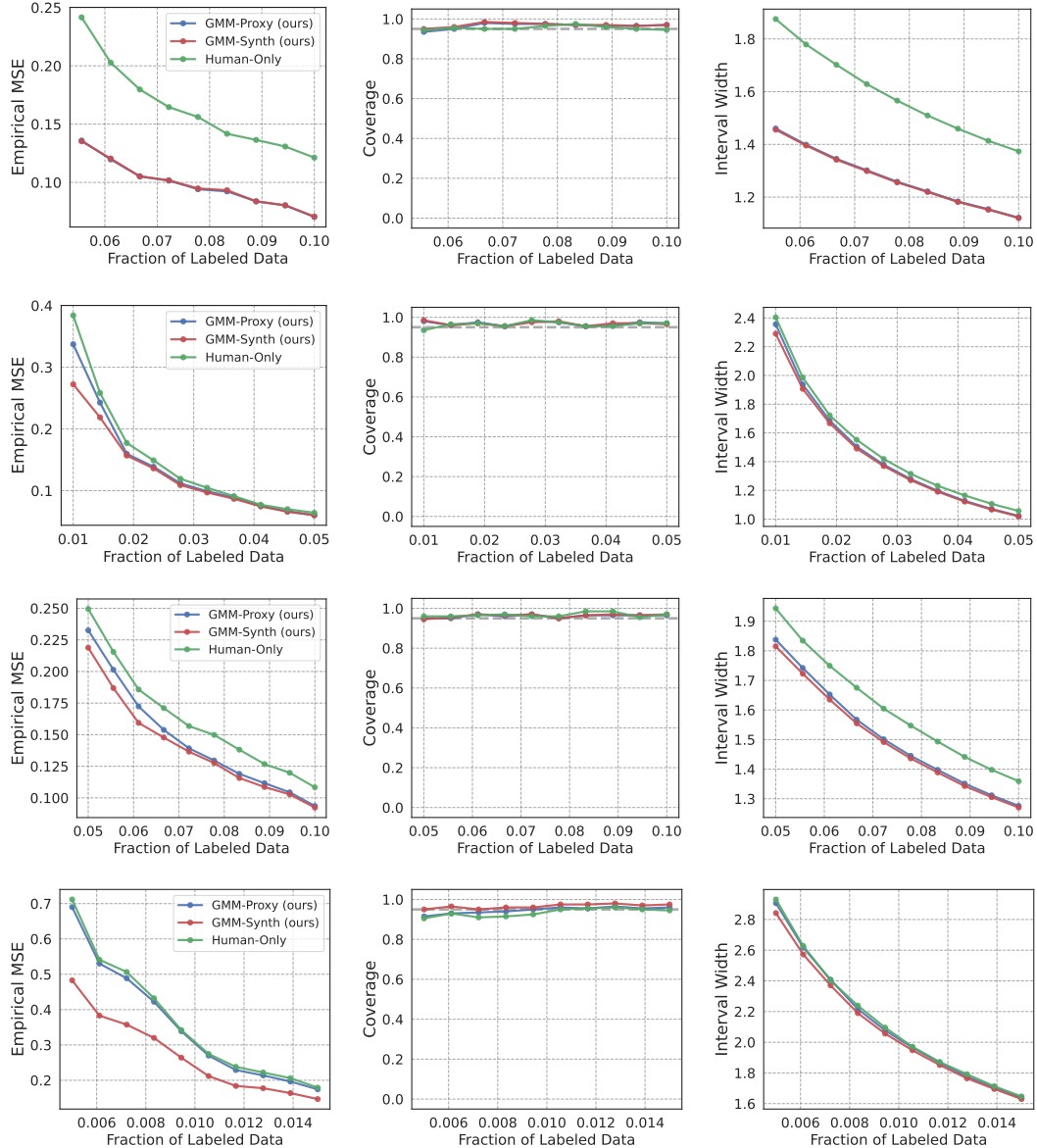

Figure 14: Zero-shot prompting results. As an ablation, we use zero-shot prompts containing the task description (instead of giving an in-context real sample) for synthetic text generation (i.e., generating $\tilde{T}$). In all tasks, we observe that our method is at least as good as Human-Only in terms of MSE and interval width. We see that the benefits to using synthetic data is much less pronounced than when using the specific sampling strategy we propose, as expected. Each row corresponds to a logistic regression task (i.e., 1pp, Hedging, Stance, Congressional Bills Data (from top to bottom)); each column corresponds to a metric (i.e., MSE, coverage, confidence interval width (from left to right)). Results are averaged over 200 trials.

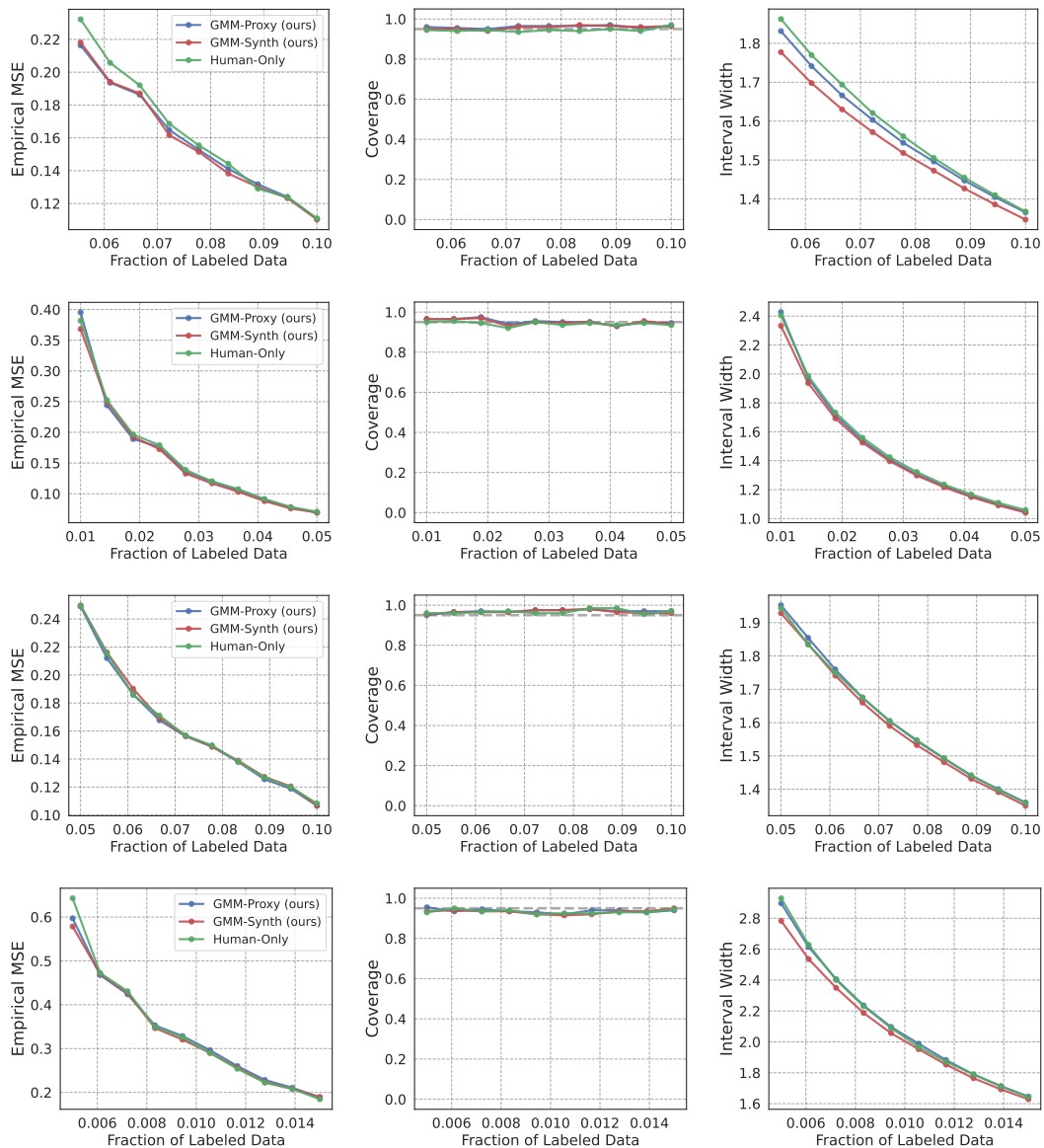

Figure 15: Random noise for labels results. As another ablation, we replace the proxy and synthetic labels ($\hat{Y}$ and $\tilde{Y}$, respectively) with random noise. Importantly, we observe that our method still performs at least as good as just having used the real data (Human-only) on all tasks. Each row corresponds to a logistic regression task (i.e., 1pp, Hedging, Stance, Congressional Bills Data (from top to bottom)); each column corresponds to a metric (i.e., MSE, coverage, confidence interval width (from left to right)). Results are averaged over 200 trials.

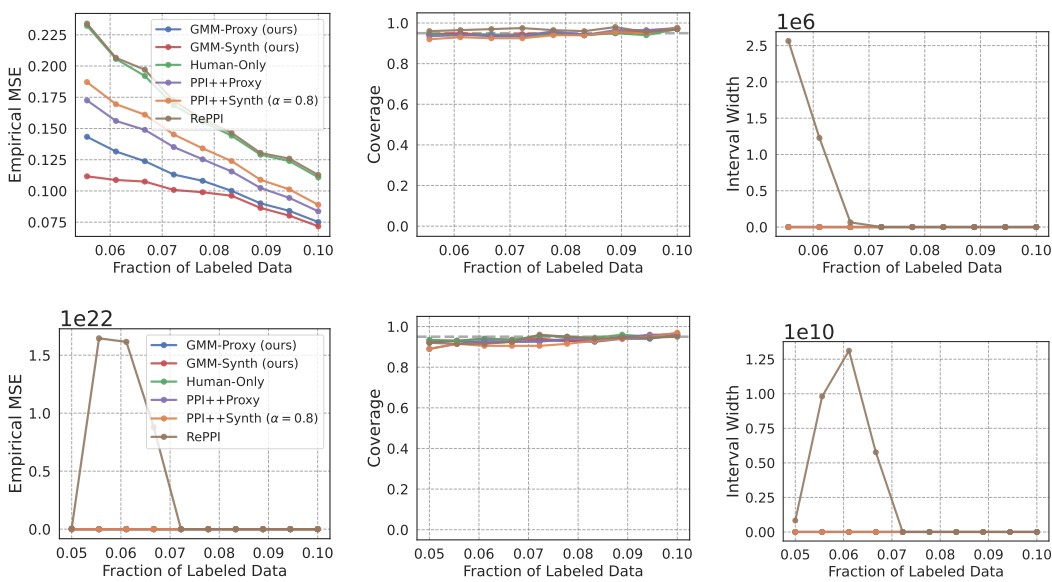

Figure 16: RePPI results for 1pp task for logistic regression (top) and OLS (bottom). We report the RePPI method results for the 1pp task separately here, due to some very large values. In the main text (Figures 1, 2), we exclude the RePPI numbers from the plots for better visibility.

