# OpenReview forum: "Valid Inference with Imperfect Synthetic Data"
_NeurIPS.cc/2025/Conference — NeurIPS 2025 poster_

### Official Review · Reviewer_B76K · 2025-06-19

**Clarity:** 2
**Significance:** 3
**Originality:** 3
**Rating:** 4
**Confidence:** 3

**Summary:**

This paper presents a novel methodology for generating synthetic data from Large Language Models (LLMs) that seeks to generate samples matching the distribution of human-labeled data. This is done by incorporating one human-labeled sample and specific request for a classification category into each LLM prompt. The paper also introduces a new estimator algorithm that seeks to correctly predict the true distribution from the (imperfect) artificial samples.

**Questions:**

Please explain why the experiments using PPI++ were conducted using a different set of artificially generated samples than the ones used by the authors' algorithm.

**Ethical Concerns:**

["Major Concern: Data quality and representativeness"]

**Final Justification:**

This paper tackles an important challenge in modern social science by proposing a method to select synthetic datasets that are guaranteed not to decrease prediction accuracies.

I am personally not convinced by the authors' assertions, but that may be due to my limited experience with the type of statistical analyses the authors are conducting. (My own expertise lies in the area of machine learning, regression models, and LLM architectures.)

**Limitations:**

Limitations are addressed briefly, but not in enough detail to be satisfactory. In particular, the authors do not address the (very real) risks that researchers acting based on the claims of the paper might attribute more reliability to the algorithms result than is justified. In other words, because the authors make strong claims relating to the algorithm's ability to identify the true distribution from artificially generated samples, there is a risk that researchers may succumb to overreliance on the algorithm's outputs, even when research is conducted that is outside the scope of the algorithm's optimal performance bounds (for example, when a small or unreliable language model is used, or when the specific classification requested by the prompt (e.g. "hedging") lies outside of the language model's capabilities. For this reason, the authors should clearly identify the bounds of their algorithm's validity: For which category of language models and for which types of classification problems can the authors' novel prompting method be considered accurate? How can this reliability be externally verified by other researchers? What are the potential outcomes or dangers of naive reliance upon the authors' algorithm in contexts where the algorithm does nor provide good results?

**Paper Formatting Concerns:**

- PPI++ is referenced within the paper and used as a key baseline, but this algorithm is not clearly explained within the text of the paper, making the comparisons difficult to understand. A more direct explanation of PPI++ within the paper body would reduce reviewer workload and make the paper more accessible to a broad audience.
- The title is a poor representation for the concepts conveyed within the paper body. "Using Imperfect Synthetic Data in Downstream Inference Tasks" gives the appearance that the authors' main contribution is the idea of using synthetic data to improve inference, a concept which is far from novel. The authors' actual contributions are far more interesting, but neither the title nor abstract clearly convey the ideas behind the authors estimator algorithm, nor the principled sampling method used by the authors to generate synthetic data.

**Quality:**

3

**Strengths And Weaknesses:**

Strengths:

- The paper seeks to address an important limitation in many social science contexts: Hand-labeled training data is expensive to produce and often limited in scope. Artificially-generated samples offer one way to overcome this limitation, but only if it is possible to infer the true distribution from an analysis of artificially generated samples (which are likely to contain unwanted biases)
- The paper includes theoretical grounding for the proposed methodology, building upon established mathematical precedents
- The authors ran multiple trials of all experiments, strengthening the reliability of the results
- The introduction of a principled sampling method for artificial generated within the context of social science research is particularly valuable, and is worthy of deeper exploration

Weaknesses
- The prompting methods used by the researchers to generate artificial samples may not be as robust as the paper implies.
- The assumption that prompting with T will result in text generations that exactly match the original distribution is incorrect, and in particular is highly dependent on the language model used. Highly distilled language models, in particular, tend to gravitate toward certain styles and patterns of generated text regardless of the prompt, and there is no guarantee that they will match T_i in all meaningful senses.
- The authors do not address the question of how the selection of language model (e.g. GPT-4, GPT-4o, Qwen, Claude Sonnet, Llama, etc.) might impact the reliability of results, nor does the work explore the question of what will happen when the specific classification requested in the LLM prompt (e.g. hedging) represents a skill, behavior, or category that the language model is not equipped to- PPI++ is referenced within the paper and used as a key baseline, but this algorithm is not clearly explained within the text of the paper, making the comparisons difficult to understand. A more direct explanation of PPI++ within the paper body would reduce reviewer workload and make the paper more accessible to a broad audience. identify or generate.
- The baseline comparisons used by the paper do not seem to be the correct way to evaluate this work. For example, PPI++ is used as a baseline, but it differs from the authors' methods along multiple dimensions (e.g. power tuning, the sampling method used to generate artificial data), making it difficult to draw meaningful conclusions from the results. In particular, it would have been nice to see a comparison of the authors' methods and PPI++ using the authors' novel sampling method for artificial data. It would also have been nice to see a comparison of PPI++ performance both with naively sampled artificial data and with data sampled using the authors' methodology.
- The presentation of ideas within the paper do not flow naturally. Motivation for the research is not clearly established, many key points of information are not included in the paper proper and must be researched elsewhere in order for the paper to be properly understood. The title and abstract do not effectively convey the main contributions of the paper.

---

> ### Author Rebuttal · Authors · 2025-07-31
>
> Dear Reviewer B76K,
>
> Thank you for your feedback! We address your concerns below:
>
> ### **Assumption that text generations = original distribution**
>
> > The prompting methods used by the researchers to generate samples may not be as robust as the paper implies. The assumption that prompting with T will result in text generations that exactly match the original distribution is incorrect
>
> We sincerely apologize for the potential confusion! To start, we must clarify: **our method does not require this assumption**. We have edited the Introduction (lines 45-48) to make this explicit (as we think the misunderstanding may be our fault due to some ambiguous wording). Changes are bolded:
>
> “””
> Naively…impossible to provide statistical guarantees for the resulting estimate if the generative model does not perfectly match the real distribution—which is expected in practice. **We propose** a specific sampling strategy…This correlation structure will prove critical for principled methods for integrating synthetic data, as it enables us to more effectively share information across them **without requiring the distributions to be equal to each other**.
> “””
>
> More importantly, we would like to highlight that **in the Methods section we do not impose any assumptions on the quality of proxy or synthetic data to derive our key theoretical results on consistency** (Section 4.4) (see Appendix A for proof). In Section 4.5, we explicitly analyze why synthetic data improves performance even when they are imperfect and derive an explicit formula for the asymptotic variance in Appendix C, where we show, “in the worst case, where synthetic data is completely informative, including it does not hurt, asymptotically” (Lines 257-258).
>
> The core idea behind how we construct our GMM estimator (see Section 4.2) is that we estimate **separate** parameters ($\theta, \eta$) for each data source, precisely how the method avoids assuming the samples (from different data sources) come from the same distribution. If we assumed all data came from the same distribution (as the original distribution) then we would constrain $\theta = \eta$. The fact that we keep these separate is precisely how we allow for distributional differences while still benefitting from cross correlations.
>
> >  “because the authors make strong claims relating to the algorithm's ability to identify the true distribution from artificially generated samples, there is a risk that researchers may succumb to overreliance on the algorithm's outputs. For this reason, the authors should clearly identify the bounds of their algorithm's validity…”
>
> We apologize for the potential confusion! We would like to clarify again that we do not claim to identify the true distribution from artificially generated samples. Our theoretical results guarantee valid confidence intervals *regardless* of whether the synthetic samples match the real distribution; the confidence intervals will just be tighter if the synthetic samples match more closely. We state explicitly in the paper, even if the LLM is not good, the performance will still be at least as good as if the researcher had only used the human-labeled data: “When the [real and synthetic] residuals are independent, the formula reduces to the optimal variance based only on the real data” (Lines 256-257).
>
> To further demonstrate this, we have added new experiments where the synthetic samples from the LLM are replaced by random noise with a set probability $p$ to further demonstrate how our method would empirically perform in even worse cases. More concretely, when p=0.5, we default to **complete random noise** for 50% of the samples and use the LLM predictions for the rest. We observe that even though the synthetic data is complete random noise for half of the samples, we don’t perform worse than not having used them (approximates the Human-Only baseline). We theoretically provide an explanation and proof in Section 4.5 (Lines 257-258) and Appendix C (Lines 831-851).
>
> |                    | 1pp   | Hedging | Stance | CBP |
> |--------------------|-------|---------|--------|---------------------|
> | Human-Only         | 0.301 | 0.384   | 0.250  | 0.566               |
> | GMM-Synth (Ours)   | 0.300 | 0.374   | 0.248  | 0.567               |
>
>
> > Ethical Concerns: Data quality and representativeness
>
> Again, we would like to clarify that we do not make claims on data representativeness and one of the main strengths of our method is specifically that it does not make assumptions on the quality of the synthetic data.
>
>
> ### **Different LLMs**
>
> > how the selection of language model might impact the reliability of results
>
> We thank the reviewer for their valuable suggestion! We have added additional experiments using small, open-source models with worse performance (Llama-3-8b, Qwen-3-8b), where we observe that even with worse quality LLMs the same conclusions hold.
>
> We report performance (MSE; lower is better) with Llama-3-8b, Qwen-3-8b:
>
> |                    | 1pp   | Hedging | Stance | CBP |
> |--------------------|-------|---------|--------|---------------------|
> | Human-Only         | 0.301 | 0.384   | 0.250  | 0.566               |
> | GMM-Synth (Ours)   | 0.227 | 0.339   | 0.238  | 0.508               |
>
> |                    | 1pp   | Hedging | Stance | CBP |
> |--------------------|-------|---------|--------|---------------------|
> | Human-Only         | 0.301 | 0.384   | 0.250  | 0.566               |
> | GMM-Synth (Ours)   | 0.198 | 0.329   | 0.229  | 0.506               |
>
> ### **Details about Baseline Comparisons**
>
> > baseline comparisons do not seem to be the correct way to evaluate this work. PPI++ is used as a baseline, but it differs from the authors' methods along multiple dimensions (e.g. power tuning, the sampling method used to generate artificial data)
>
> We would like to clarify that **PPI++ uses the same synthetic data samples**. The synthetic data samples used across all baselines are fixed to be the same across all methods.
>
> We would also like to clarify that power tuning is a strategy the PPI++ authors use to improve their original weaker baseline PPI. Power tuning finds an optimal weighting parameter $\lambda$ that takes a combination between the classical estimator (using only real data) and the PPI estimator (using both real and synthetic data). $\lambda$ is chosen to minimize the width of the confidence interval. When the prediction model performance is poor, such that incorporating unlabeled data hurts performance, it sets $\lambda$ = 0 to recover the classical estimator. Therefore, we are actually making a comparison with a stronger method by reporting numbers that involve power tuning for that baseline. We also report numbers without power tuning for more context:
>
> |                    | 1pp   | Hedging | Stance | CBP |
> |--------------------|-------|---------|--------|---------------------|
> | PPI++Synth         | 0.347 | 0.380   | 0.227  | 0.690               |
> | PPI++Synth (without power tuning)   | 0.358 | 0.402   | 0.258  | 0.710               |
>
>
> ### **Title and Abstract**
>
> > The authors' actual contributions are far more interesting, but neither the title nor abstract clearly convey the ideas behind the authors estimator algorithm, nor the principled sampling method used by the authors to generate synthetic data.
>
> We appreciate that you find our contributions interesting! We have updated the title to “Valid Inference with Imperfect Synthetic Data via Generalized Method of Moments” to better reflect the core methodology behind the estimator algorithm and have updated the abstract to include further details about the principled sampling method, the estimator algorithm, and intuition behind why/when synthetic data improves performance.
>
>
> ### **Expand Motivation**
>
> > Motivation is not clearly established
>
> We are motivated by limited-labeled data settings where practitioners face constraints in obtaining sufficient human-labeled data due to budget constraints. This has led researchers to increasingly turn to LLMs as cheap but noisy alternatives for data generation and annotation (e.g., simulating human responses to surveys) [1][2][3].
>
> The persistent challenge is that naively combining these imperfect synthetic data samples with human-annotated samples for downstream inference tasks can lead to biased estimates, compromising statistical validity of its conclusions. Current methods are only able to incorporate noisy predictions and not fully synthetic samples [4][5], which further motivates the purpose of this work. Our work addresses this fundamental gap by providing the first theoretically-grounded framework that can safely incorporate fully synthetic samples while maintaining statistical validity, regardless of the quality of the synthetic data.
>
> ### **Add Detailed Description of PPI++ in Main Text**
>
> > PPI++ is referenced within the paper and used as a key baseline, but this algorithm is not clearly explained within the text of the paper
>
> Thanks for the suggestion! Currently, we state the PPI++ objective in Section 5, Proposition 2 (Lines 274-277) and the pseudocode for the algorithm implementation in Appendix F, Algorithm 1 (Lines 894-895). To address this concern, we have (1) updated the draft to add the pseudocode in the main text (Section 5) and (2) add a line-by-line explanation of the algorithm in Section 5 to help provide a more direct explanation of the method.
>
> We thank the reviewer again for their time and consideration! We are more than happy to address any other concerns they might have.
>
>    1. Hwang et al. Opportunities and challenges of applying llm-simulated data, 2025.
>    2. Geng et al. Are large language models chameleons? an attempt to simulate social surveys, 2024.
>    3. Rothschild et al. Opportunities and risks of llms in survey research, 2024.
>    4. Gligoric et al. Can Unconfident LLM Annotations Be Used for Confident Conclusions, 2024.
>    5. Egami et al. Using Imperfect Surrogates for Downstream Inference, 2023.

---

> > ### Comment · Reviewer_B76K · 2025-08-01
> >
> > Thank you for the clarifications. I continue to have some concerns about what may occur when models are prompted to classify behaviors that lie outside of their training distribution. (I'm not completely convinced that such instances can be effectively simulated via the 50% random noise experiment, as the noise is likely centered at 0, whereas misaligned LLM generations will likely all be biased in the same direction.)
> >
> > I think it will not be possible to allay my concerns via an online discussion alone. However, in future versions of this work, the authors are encouraged to consider what experiments or simulations can adequately demonstrate that even very poor text generations based on out-of-domain or otherwise challenging prompts will not harm the integrity of results. (My biggest concern stems from the fact that the method asserts guarantees on classification accuracy. I do not want researchers who use this method to blindly rely on results that may turn out to be incorrect.)

---

> ### Author Response · Authors · 2025-08-01
> **Response to Reviewer B76K (1/2)**
>
> Dear Reviewer B76K,
>
> Thank you for your response! Below, we respond explicitly to each point raised:
>
>   *1. We would like to clarify that our paper does not make this claim and we acknowledge this explicitly in the Limitations section of our original submission*
>
> First, we would like to remark that our original submission acknowledges this explicitly in the Limitations section of our submission (Lines 348-352):
>
> “A potential limitation of our framework is its reliance on the quality of the generative model (e.g., an LLM). As with other debiasing approaches in the literature, very poor-quality synthetic data would yield little-to-no benefits in statistical efficiency. Moreover, our theoretical guarantees, like those of debiasing methods, hold asymptotically and thus may fail to hold in extremely low-data regimes, potentially leading to undercoverage of the target parameter.”
>
>   *2. We do not assert any guarantees on classification accuracy.*
>
> We apologize for the confusion. To clarify: we never claim to guarantee “classification accuracy” in the sense of the individual classification outputs from the LLM. Rather, our guarantees are on the “consistency and proper asymptotic coverage” of statistical inferences, *regardless* of how accurate the LLMs’ classifications are (see e.g. Lines 35-36 and 72-73 in the Intro for examples of this specific phrasing). We then prove that consistency and proper asymptotic coverage hold in Proposition 4.1 in the paper. Is there a specific concern about the proof of Proposition 4.1?
>
>   *3. Current experiments in the paper actually already demonstrate the setting when LLM generations are very biased (see Figure 7)*
>
> We would like to clarify that our experiments in the paper explicitly include settings where the LLM generations are very biased (Please see Figure 7 where we plot bias). In Figure 7, we can clearly observe that using the LLM generations alone leads to largely biased estimates that are skewed in one direction. In Figure 3 (Main Results), we observe the performance of our method using these *same* LLM generations.

---

> ### Author Response · Authors · 2025-08-01
> **Response to Reviewer B76K (2/2)**
>
> *4. We add additional experiments encompassing (1) noise in one direction, (2) anticorrelation, (3) worse quality LLMs, (4) and worse quality prompting strategies*
>
> We want to thank Reviewer B76K for these insightful suggestions! First, we would like to highlight that during the rebuttal, we have also conducted experiments using worse quality, open-source LLMs (Llama-3-8b, Qwen-3-8b) to demonstrate our method’s performance with worse quality synthetic data, as requested (e.g., the accuracy of predictions from Llama-3-8b is only 0.15 and for Qwen-3-8b is 0.23 for the Hedging task demonstrating poor performance for the target task). Note that we employ all LLMs in a zero-shot manner with no domain-specific fine-tuning in all experiments.
>
> To explicitly address the setting of when all "LLM generations are biased towards one direction" we provide an experiment where instead of random samples between {0, 1}, we assign towards a single class, which reflects your point about how misaligned LLM generations can be biased in one direction.
>
> |                    | 1pp   | Hedging | Stance | CBP |
> |--------------------|-------|---------|--------|---------------------|
> | Human-Only         | 0.301 | 0.384   | 0.250  | 0.566               |
> | GMM-Synth (Ours)   | 0.302 | 0.379   | 0.246  | 0.565               |
>
> To further address your concern, we also add an experiment where the LLM label is anticorrelated with the real label.
>
> |                    | 1pp   | Hedging | Stance | CBP |
> |--------------------|-------|---------|--------|---------------------|
> | Human-Only         | 0.301 | 0.384   | 0.250  | 0.566               |
> | GMM-Synth (Ours)   | 0.116 | 0.132   | 0.109  | 0.187               |
>
> To further address your concern, we also add experiments that use varying types of simpler prompt strategies: few-shot prompting and zero-shot prompting, which also result in worse quality synthetic data.
>
> Results for few-shot prompting, zero-shot prompting (respectively):
>
> |  | 1pp | Hedging | Stance | CBP |
> |----------------------|----------------|---------------------|--------|---------------------|
> | Human-Only           | 0.301          | 0.384               | 0.250  | 0.566               |
> | GMM-Synth (Ours)     | 0.246          | 0.321               | 0.209  | 0.497              |
>
>
> |  | 1pp | Hedging | Stance | CBP |
> |----------------------|----------------|---------------------|--------|---------------------|
> | Human-Only           | 0.301          | 0.384               | 0.250  | 0.566               |
> | GMM-Synth (Ours)     | 0.277          | 0.370              | 0.239  | 0.507               |
>
>
> We sincerely thank Reviewer B76K again for their constructive remarks and suggestions to further enhance the impact of our work. We believe the following changes will address the reviewer’s concerns:
>
> - Add further explicit statements in the Introduction and Limitations section to clarify guarantees and guide practitioners for correct usage of the method.
> - Add results on random noise, noise in one direction, anticorrelation, worse-quality LLMs, worse quality prompting strategies to Section 6 to empirically evaluate our method in settings with poor LLM performance with both simulations and real data.
> - Add a discussion of Figure 7 into the main text, which demonstrates the setting where the LLM is very biased.
>
> We hope these revisions address Reviewer B76K's concerns and improve our manuscript to meet their expectations for an acceptable article. We are more than happy to address any other concerns they might have.

---

> ### Comment · Reviewer_B76K · 2025-08-06
>
> Thank you for the additional information, and for the reference to Figure 7 in the Appendix. I have revised my score upward.

---

> ### Author Response · Authors · 2025-08-07
>
> Thanks! We are delighted to hear that our rebuttal was helpful in addressing your concerns! Best wishes

---

### Official Review · Reviewer_WjYv · 2025-07-03

**Clarity:** 1
**Significance:** 3
**Originality:** 3
**Rating:** 3
**Confidence:** 2

**Summary:**

Using human annotators to label domain-specific text can be highly expensive and time-consuming. Therefore, recent works have used LLMs to replace humans in labeling and data generation. While this can remove the original costs, we are still not sure about the benefits and correctness of the synthetic data. This paper models this process through the Generalized Method of Moments. The authors model the contribution of real, auxiliary, and synthetic data. Their comprehensive model shows we can utilize synthetic data to improve performance under essential conditions. They should have similar residuals to real data.

**Questions:**

Conditions under line 135 are not clear. Are the equations showing how T, X, and Y are generated, or are T and X when the text is labeled, and T and Y when the text is unlabeled? The current formulation is misleading.

**Ethical Concerns:**

["NO or VERY MINOR ethics concerns only"]

**Final Justification:**

After reading the author's response to all the reviews, I decided to keep my score.

**Limitations:**

Yes

**Quality:**

3

**Strengths And Weaknesses:**

**Weakness**

The main weakness of the paper is that it is hard to follow; some basic concepts are overexplained, while the important part and the contribution section are left ambiguous. A very critical concept, the Generalized Method of Moments, is not explained in the related work.

**Strength**

The contribution of the paper is significant. The authors have theoretically shown how to select synthetic datasets and when they can be beneficial.

The experimental results also support their theoretical results.

The paper has an interesting discussion section (4.5 and 5).

My suggestion for the authors is to summarize the content in the early pages of the paper and expand on how the Generalized Method of Moments works and how it can help with this problem.

---

> ### Author Rebuttal · Authors · 2025-07-31
>
> Dear Reviewer WjYv,
>
> Thank you for your feedback! We’re grateful for your acknowledgement of our contributions. We address your concerns below:
>
> ### **Clarity and Expansion of Related Work**
>
> > The contribution of the paper is significant. The main weakness of the paper is that it is hard to follow; some basic concepts are overexplained, while the important part and the contribution section are left ambiguous. A very critical concept, the Generalized Method of Moments, is not explained in the related work.
>
> We appreciate that you find our contributions to be significant and thank you for this important feedback. We currently discuss our GMM approach in detail in the Methods section (Section 4). To address this further, we add the following subsection to the Related Work section (after Line 104) to provide further context earlier on in the draft:
>
> “””
> **Generalized Method of Moments (GMM).** The GMM framework [Hansen, 1982] is a flexible estimation method that generalizes many common statistical approaches (e.g., maximum likelihood, instrumental variables, least squares, etc.). GMM estimators are defined by moment conditions—functions of the data and parameters whose expectations equal zero at the true parameter values. The key insight is that when the number of moment conditions exceeds the number of parameters to estimate, GMM optimally combines these conditions using a weighting matrix that accounts for their variance-covariance structure. This framework has been widely adopted in statistics and econometrics due to its robustness to distributional assumptions and ability to handle overidentified systems (more conditions than parameters). In this work, we leverage GMM's flexibility to incorporate human-labeled, proxy, and synthetic data through separate moment conditions. The two-step GMM estimation procedure [Newey and McFadden, 1994] (see Section 4.3 for details) learns optimal weights for combining these data sources through the moment covariance matrix, eliminating manual hyperparameter tuning required by existing debiasing methods. The key intuition is that the weighting matrix captures the interactions between the real moment residuals and the synthetic moment residuals; the interactions between these moment residuals improves estimation when they are predictive of one another (see Section 4.5 for further discussion).
> “””
>
>
> Currently, we provide a description of our GMM method in Section 4 in the following manner:
> Introduce moment conditions (Section 4.1), estimator construction with proxy and synthetic data (Section 4.2), the two-step estimation procedure (Section 4.3), and theoretical results on consistency and asymptotic behavior of our GMM estimator (Section 4.4 with the full proof in Appendix A).
>
> We further include a concrete example of our moment construction in Appendix B (Lines 821-831). However, we forgot to note a reference to Appendix B in the main text, which might have caused some ambiguity. To help improve clarity, we have updated the draft to move this concrete example from Appendix B into the main text before Section 4.3. This provides readers with a concrete illustration of our moment construction before turning to the estimation procedure (Section 4.3), making the method easier to follow. We hope these additions help step through the main contribution/method of the paper better. We would be happy to further improve the explanation if there are any additional remaining concerns.
>
> > My suggestion for the authors is to summarize the content in the early pages of the paper and expand on how the Generalized Method of Moments works and how it can help with this problem.
>
> Thank you for this suggestion! We have updated the Introduction (Lines 61-70) with the following paragraph to summarize and further expand on how the GMM works and helps with the problem at hand more earlier in the draft:
>
> """
> Our primary methodological contribution is to propose a new estimator based on the generalized method of moments (GMM) framework that naturally incorporates this synthetic data. The construction of our GMM estimator defines separate parameters and moments for each data source. It is not initially obvious whether the incorporation of moments based exclusively on synthetic data should yield any benefits (or even affect) the estimation of the target parameter. However, we find that in the GMM estimation procedure, the off-diagonal terms in the weight matrix captures interactions between the real moment residuals and synthetic moment residuals, allowing the information from the synthetic data moments to influence the estimation of the target parameter. The key intuition is that synthetic data will improve performance when the synthetic data residuals are predictive of the real data residuals. More concretely, the asymptotic variance is the remaining variance after you regress the real residuals on the span of the synthetic residuals. In other words, when the residuals are independent of each other, the asymptotic variance reduces to the optimal variance based only on real data. This means that in the worst case, when synthetic data is completely uninformative, it does not hurt performance asymptotically (see Section 4.5 and Appendix C for further details).
> """
>
>
> ### **Clarification Question (Line 135)**
>
> > Conditions under line 135 are not clear. Are the equations showing how T, X, and Y are generated, or are T and X when the text is labeled, and T and Y when the text is unlabeled? The current formulation is misleading.
>
> Thank you for raising this clarification point! Yes, the equations are showing how T,X,Y are generated. We have updated the draft to list the equations in a list and not side-by-side to avoid this potential confusion:
>
> $
> \tilde{T}_k \sim \mathbb{P} (\cdot \mid T_i, X_i)
> $
> if labeled
>
> $
> \tilde{T}_k \sim \mathbb{P} (\cdot \mid T_j, \hat{X}_j)
> $
> if unlabeled
>
> $
> \tilde{X}_k \sim \mathbb{P} (\cdot \mid \tilde{T}_k)
> $
>
> $
> \tilde{Y}_k \sim \mathbb{P} (\cdot \mid \tilde{T}_k)
> $
>
> We thank the reviewer again for their time and consideration! We are more than happy to address any other concerns they might have.

---

> ### Author Response · Authors · 2025-08-07
> **Hope to hear back soon**
>
> Dear Reviewer WjYv,
>
> We thank the reviewer for their comments that they find our contributions to be significant and that our empirical evaluations support our theoretical results. In our response above, we have tried to address all your questions and comments. According to your suggestions, we have strengthened our explanation of Generalized Method of Moments by adding a new section to the draft, moved up a concrete example from Appendix B into the main text, and summarized the content on how the GMM works and how it helps with the problem in early pages of the paper, as suggested.
>
> We once again thank you for your valuable time spent reviewing our work. We hope these revisions address Reviewer WjYv's concerns and improve our manuscript to meet their expectations for an acceptable article. We are more than happy to address any other concerns they might have.

---

### Official Review · Reviewer_psyf · 2025-07-07

**Clarity:** 3
**Significance:** 3
**Originality:** 3
**Rating:** 4
**Confidence:** 3

**Summary:**

This paper proposes a principled framework for incorporating fully synthetic data, generated by LLMs, into downstream statistical inference tasks. Motivated by applications in computational social science where labeled data is often scarce, the authors introduce a novel estimator based on the Generalized Method of Moments that combines labeled data, model-predicted data (proxies), and LLM-generated synthetic samples. The method exploits correlation structures introduced by conditioning synthetic samples on real data and provides theoretical guarantees on consistency and asymptotic normality. Empirical results on four real-world regression tasks demonstrate significant performance gains in low-label regimes.

**Questions:**

1. Can your framework support generation strategies that are not conditioned on real examples? For instance, can it be extended to few-shot or zero-shot prompting?

2. What's the perfomance gap of between open-source models and GPT-4o?

3. What's the diversity of the generated samples, which should also be dicsussed and verified.

**Ethical Concerns:**

["NO or VERY MINOR ethics concerns only"]

**Final Justification:**

After rebuttal, my concerns are properly addressed. Based on the paper quality and overall concerns, I will keep my score unchanged which lean to accept.

**Limitations:**

Yes

**Quality:**

2

**Strengths And Weaknesses:**

**Strengths**

1. This paper tackles an important and emerging challenge, i.e., how to use LLM-generated synthetic data for reliable statistical inference.

2. It introduces GMM estimator for this task and leverages the correlation between real and synthetic data for synthetic data generation. Experiments on diverse real-world tasks from political science and psychology show strong gains, especially in low-data settings.

3. The paper is well-written and clearly structured. And the theoretical and empirical sections are well balanced.

**Weaknesses**

1. While the paper mentions that low-quality synthetic data won’t help, it doesn’t empirically explore the effect of varying LLM quality or prompt design.

2. The experiments are focused on regression tasks with structured covariates; it’s unclear how well the framework generalizes to classification, causal inference, or time series.

3. The framework relies on the availability of strong LLMs like GPT-4o. This may not generalize well to low-resource settings, such as open-source models.

4. It would be helpful to see a comparison between the proposed conditioned sampling vs. naive zero-shot generation to justify the design decision.

---

> ### Author Rebuttal · Authors · 2025-07-31
>
> Dear Reviewer psyf,
>
> Thank you for your feedback! We address your concerns below:
>
> ### **Varying LLM quality/prompt design types**
>
> > While the paper mentions that low-quality synthetic data won’t help, it doesn’t empirically explore the effect of varying LLM quality…The framework relies on the availability of strong LLMs like GPT-4o. This may not generalize well to low-resource settings, such as open-source models.
>
> Thanks for the suggestion! We have added additional experiments using small, open-source models (Llama-3-8b, Qwen-3-8b) to encompass varying types of LLM quality. We observe that even with worse quality LLMs, the same conclusions hold. We have also added an experiment where the synthetic samples from the LLM are replaced by random noise with a set probability $p$ to further demonstrate how our method would empirically perform in ***even worse cases***. More concretely, when p=0.5, we default to **complete random noise** for 50% of the samples and use the LLM predictions for the rest. We observe that even though the synthetic data is complete random noise for half of the samples, our method is able to safeguard against this (approximates performance you would get with human-only samples). This is important since practitioners can only know retrospectively if an LLM was not good for their task at hand. We would like to highlight that we theoretically provide an explanation and proof of this in Section 4.5 (Lines 257-258) and Appendix C (Lines 831-851).
>
> We report performance (in terms of MSE; lower is better) with Llama-3-8b, Qwen-3-8b (respectively)
>
> |                    | 1pp   | Hedging | Stance | CBP |
> |--------------------|-------|---------|--------|---------------------|
> | Human-Only         | 0.301 | 0.384   | 0.250  | 0.566               |
> | GMM-Synth (Ours)   | 0.227 | 0.339   | 0.238  | 0.508               |
>
> |                    | 1pp   | Hedging | Stance | CBP |
> |--------------------|-------|---------|--------|---------------------|
> | Human-Only         | 0.301 | 0.384   | 0.250  | 0.566               |
> | GMM-Synth (Ours)   | 0.198 | 0.329   | 0.229  | 0.506               |
>
> For 50% of the samples, we default to complete random noise.
>
> |                    | 1pp   | Hedging | Stance | CBP |
> |--------------------|-------|---------|--------|---------------------|
> | Human-Only         | 0.301 | 0.384   | 0.250  | 0.566               |
> | GMM-Synth (Ours)   | 0.300 | 0.374   | 0.248  | 0.567               |
>
> We would also like to remark that our method offers a statistical framework for integrating information from LLMs, which provides value when LLMs offer useful signal (as you should only expect it to help if the LLM is “good”). In the case when the LLM is “bad”, what is important is that the method does not result in extremely biased estimates and can safeguard against this. The performance of models today and their rapidly improving accessibility (e.g., open models also keep improving) is what motivates practitioners to want to leverage predictions and generations from them, which in turn provides strong motivation for such integration frameworks.
>
> > Varying prompt design. It would be helpful to see a comparison between the proposed conditioned sampling vs. naive zero-shot generation to justify the design decision…Can your framework support generation strategies that are not conditioned on real examples? For instance, can it be extended to few-shot or zero-shot prompting?
>
> We thank the reviewer for their valuable suggestion. We have added additional experiments encompassing varying types of prompt strategies: few-shot prompting and zero-shot prompting. We clearly see that conditional sampling leads to larger benefits downstream, justifying our design decision. However, even when the synthetic samples are of worse quality due to simpler prompt strategies, we still outperform baselines.
>
> Results for few-shot prompting, zero-shot prompting (respectively):
>
> |  | 1pp | Hedging | Stance | CBP |
> |----------------------|----------------|---------------------|--------|---------------------|
> | Human-Only           | 0.301          | 0.384               | 0.250  | 0.566               |
> | GMM-Synth (Ours)     | 0.246          | 0.321               | 0.209  | 0.497              |
>
>
>
> |  | 1pp | Hedging | Stance | CBP |
> |----------------------|----------------|---------------------|--------|---------------------|
> | Human-Only           | 0.301          | 0.384               | 0.250  | 0.566               |
> | GMM-Synth (Ours)     | 0.277          | 0.370              | 0.239  | 0.507               |
>
>
> ### **More Tasks**
> > The experiments are focused on regression tasks with structured covariates; it’s unclear how well the framework generalizes to classification, causal inference, or time series.
>
> We have added additional experiments to now cover regression, classification, causal inference, and time series tasks. We report performance in terms of MSE (lower is better) averaged over 200 trials. We observe that our method outperforms baselines across the additional tasks, demonstrating our framework's flexibility beyond regression: any estimand expressible as a moment condition can leverage our approach.
>
> First, in addition to the logistic regression results (classification task) in the paper, we run ordinary least squares (regression task). We clearly see that our method outperforms baselines and the same conclusions hold.
>
> |                    | 1pp   | Hedging | Stance | CBP |
> |--------------------|-------|---------|--------|---------------------|
> | Human-Only         | 0.022 | 0.021   | 0.036  | 0.034               |
> | PPI++Synth         | 0.020 | 0.022   | 0.035  | 0.026               |
> | GMM-Synth (Ours)   | 0.009 | 0.015   | 0.022  | 0.015               |
>
> Next, we construct a semi-synthetic causal inference task to evaluate treatment effect estimation under missing covariates. Starting with real data containing features $X_i$ and outcomes $Y_i$, we synthetically generate treatment assignments $D_i$ via covariate-dependent randomization and simulate causal outcomes with heterogeneous treatment effects across covariate strata.
> We define moment conditions $m_i(\tau) = S_i · (\phi_i - \tau)$, where $\phi_i$ is the augmented inverse propensity weighted (AIPW) influence function combining outcome regression and propensity score components. Nuisance functions (propensity scores and outcome models) are estimated non-parametrically on labeled data, then we use these moments in our GMM estimator.
> This demonstrates our framework's flexibility beyond regression: any estimand expressible as a moment condition can leverage our approach.
>
> | sel_prob | gmm-synth | gmm-proxy | ppi-synth | ppi-proxy | human-only |
> |----------|-----------|-----------|-----------|-----------|-------------|
> | 0.01     | 0.013897  | 0.014038  | 0.013891  | 0.014437  | 0.022648    |
> | 0.02     | 0.006132  | 0.006223  | 0.006490  | 0.006292  | 0.008589    |
> | 0.03     | 0.004239  | 0.004302  | 0.004402  | 0.004293  | 0.006024    |
> | 0.04     | 0.003213  | 0.003352  | 0.003300  | 0.003339  | 0.005192    |
> | 0.05     | 0.002316  | 0.002411  | 0.002484  | 0.002432  | 0.004100    |
>
> We further demonstrate our applicability to time series by estimating autoregressive (AR) parameters under missing data. We generate $N$ independent AR(1) processes $Y_{it} - \mu = \phi(Y_{i,t-1} - \mu) + \varepsilon_{it}$ where $\phi$ is the target parameter, supplemented by noisy proxy and auxiliary time series data. The GMM formulation uses AR(1) moment conditions: $E[\varepsilon_{it}] = 0$ and $E[(Y_{i,t-1} - \mu)·\varepsilon_{it}] = 0$, where $\varepsilon_{it}(\phi,\mu)$ represents model residuals. Again, we observe that our method outperforms baselines.
>
> | sel_prob | gmm-synth | gmm-proxy | ppi-synth | ppi-proxy | human-only |
> |----------|-----------|-----------|-----------|-----------|-------------|
> | 1        | 0.01053   | 0.01161   | 0.01387   | 0.01226   | 0.05426     |
> | 2        | 0.00421   | 0.00464   | 0.00548   | 0.00471   | 0.02714     |
> | 3        | 0.00257   | 0.00293   | 0.00313   | 0.00274   | 0.01645     |
> | 4        | 0.00194   | 0.00225   | 0.00246   | 0.00229   | 0.01062     |
> | 5        | 0.00146   | 0.00176   | 0.00198   | 0.00182   | 0.00935     |
>
>
> ### **Questions**
>
> > What's the performance gap between open-source models and GPT-4o?
>
> We report the performance gap between GPT-4o and open-source models (Llama-3-8b, Qwen-3-8b) in the table below.
> |                                           | GPT-4o | Llama-3-8b | Qwen-3-8b |
> |-------------------------------------------|--------|------------|-----------|
> | Hedging (Covariate Prediction) | 0.764  | 0.485      | 0.501     |
> | Hedging (Label)     | 0.785  | 0.150      | 0.230     |
> | 1pp (Covariate)     | 0.983  | 0.878      | 0.866     |
> | 1pp (Label)         | 0.785  | 0.150      | 0.230     |
> | Stance (Covariate)             | 0.864  | 0.620      | 0.645     |
> | Stance (Label)                 | 0.742  | 0.636      | 0.621     |
> | CBP (Covariate)| 0.125  | 0.087      | 0.098     |
> | CBP (Label)    | 0.827  | 0.558      | 0.557     |
>
> > What's the diversity of the generated samples, which should also be discussed and verified.
>
> Thank you for this suggestion! We report Self-BLEU (lower is more diverse) and embedding-based metrics (average cosine distance) (higher is more diverse). We also report the metrics for samples generated by different prompt strategies (zero-shot, few-shot) for comparison.
>
> |                         | Self-BLEU | Embedding-based Diversity |
> |-------------------------|-----------|---------------------------|
> | Conditioned Sampling (Ours) | 0.324     | 0.687                     |
> | Few-Shot Sampling          | 0.8599    | 0.407                     |
> | Zero-Shot Sampling         | 0.6245    | 0.504                     |
>
> We thank the reviewer again for their time and consideration! We are more than happy to address any other concerns they might have.

---

> > ### Comment · Reviewer_psyf · 2025-08-07
> >
> > Thanks for your response which has addressed my concerns. As my score already leans toward acceptance, I will keep it upchanged.

---

> ### Author Response · Authors · 2025-08-07
>
> Thank you for confirming that our response has addressed all your concerns! We sincerely appreciate your valuable time and feedback. Best wishes

---

### Official Review · Reviewer_7qA1 · 2025-07-07

**Clarity:** 3
**Significance:** 2
**Originality:** 3
**Rating:** 4
**Confidence:** 1

**Summary:**

The paper aims to find regression coefficients based on mostly-unlabeled data. It introduces `GMM+synth`, which relies on a black-box LLM to augment unlabeled data, and generate fully synthetic data. It uses a Generalized Methods of Moments approach for combining real, proxy, and synthetic data to get statistical estimates of the regression coefficients.

**Questions:**

See weaknesses. Additionally:

1. In L.135, do you mean $\tilde{X}_k=f(\tilde{T}_k)$, idem ditto for $Y$?
2. If we have enough texts $T$, what is the intuition for why modelling synthetic $T$ helps?

**Ethical Concerns:**

["NO or VERY MINOR ethics concerns only"]

**Final Justification:**

My initial concerns were about the somewhat limited exploration of some of the modelling choices (e.g. LLM choice) and empirical results. The authors have mostly resolved these issues, and I hope they will include these additional experiments in the revised manuscript. I retain my score and lean towards acceptance.

**Limitations:**

Yes

**Quality:**

2

**Strengths And Weaknesses:**

## Strengths
1. Robust low sample size inference is an underexplored problem in ML
2. The approach---building statistically valid estimators for incorporating possibly erroneous LLMs for augmentation---is appealing
3. Writing is generally good

## Weaknesses
I have two general concerns
1. Method. The method is rather involved, with ML model $f$ giving proxies, the LLM giving synthetic data, and then a GMM doing the statistical inference. There are many hidden choices---which ML model $f$ for proxy data? Which LLM for synthetic data? Which LLM prompts?---which the writers do not explore. To me as a reader, I'm not convinced the method is the best way to leverage the internal knowledge of an LLM.
2. Experiments. Related to the above, the experiments are rather limited.
   1. There is just one baseline
   2. Tasks are simple.
   3. Just a single LLM, and unclear how accurate GPT-o is for these tasks. The authors pose in Section 4.5 that we'll never be worse, as the model learns to ignore the synthetic data if it's uncorrelated. It would have been interesting to test this in practice, either with more difficult tasks or worse LLMs.

### Minor
1. Key observations in Experiments should be structured better. Right now, it jumps from Figures 3 and 4, to Figure 6 in the Appendix, to finally get to Figure 2, but then going back to Figure 3.

---

> ### Author Rebuttal · Authors · 2025-07-31
>
> Dear Reviewer 7qA1,
>
> Thank you for your feedback! We address your concerns below:
>
> ### **Method details**
>
> > The method is rather involved, with ML model giving proxies, the LLM giving synthetic data, and then a GMM doing the statistical inference.  which ML model for proxy data? Which LLM for synthetic data? Which LLM prompts?
>
> We apologize for the confusion! The **same model** (same LLM) is used to give proxy data (predictions) and synthetic data (generations). We have added the following clarification in the Methods section (Line 126) to clarify this point: “For simplicity, we use the same machine learning $f$ used above for proxy data also for synthetic data generation.”
> For the LLM, we use GPT-4o (Line 303). For the LLM prompts, we include the full prompts used for all prediction and generation tasks in Appendix D.1 (Lines 853-860).
>
> ### **Ablations and more baselines**
>
> > There is just one baseline
>
> We would like to highlight that PPI++ is the current state-of-the-art method for these tasks. This line of work termed prediction-powered inference (also referred to as debiasing-based methods) has been the predominant strategy for incorporating model predictions in the literature [1][2] and represents the current most competitive method in incorporating imperfect LLM predictions [3][4]. Therefore, we focused on comparisons with this method rather than including multiple, weaker baselines from earlier iterations of this line of research, since they resulted in worse performance.
>
> We would like to further highlight that we also implement **an “oracle” version of PPI++** that “cheats” in hyperparameter selection  (this is the baseline shown in the main plots). PPI++ requires selecting a hyperparameter alpha via cross-fitting, which cuts the sample size and leads to worse performance. To obtain an upper bound on the performance of this family of methods even if the hyperparameter could be chosen perfectly, the oracle method runs PPI++ on a large grid of hyperparameters and then reports the result closest to the ground truth (which isn’t knowable in practice). Our method outperforms even this strong baseline, while Appendix E (Fig 6) shows that the real, cross-fitting version of PPI++ performs worse than the oracle, as expected.
>
> To better address your concern, we have also **added an additional baseline** Ji et. al [5]. We report the MSE (lower is better) averaged over 200 trials in the following Table. We clearly see that our method outperforms this method and the same conclusions hold.
>
> |                    | 1pp | Hedging | Stance | CBP |
> |--------------------|----------------|---------------------|--------|---------------------|
> | Ji et al. [2]      | 0.225          | 0.387               | 0.230  | 0.740               |
> | GMM-Synth (Ours)   | 0.125          | 0.162               | 0.208  | 0.311               |
>
>
> > Just a single LLM, and unclear how accurate GPT-o is for these tasks. It would have been interesting to test [worse synthetic data] in practice, either with more difficult tasks or worse LLMs.
>
> We thank the reviewer for their valuable suggestion! We have added additional experiments using small, open-source models  (Llama-3-8b, Qwen-3-8b), where we observe that even with worse quality LLMs the same conclusions hold.  We have also added an experiment where the synthetic samples from the LLM are replaced by random noise with a set probability $p$ to further demonstrate how our method would empirically perform in even worse cases. More concretely, when p=0.5, we default to **complete random noise** for 50% of the samples and use the LLM predictions for the rest. We observe that even though the synthetic data is complete random noise for half of the samples, we don’t perform worse than not having used them (outperforms the Human-Only baseline). We would like to highlight that we theoretically provide an explanation and proof of this in Section 4.5 (Lines 257-258) and Appendix C (Lines 831-851).
>
> We report performance (MSE; lower is better) with Llama-3-8b, Qwen-3-8b (respectively)
>
> |                    | 1pp   | Hedging | Stance | CBP |
> |--------------------|-------|---------|--------|---------------------|
> | Human-Only         | 0.301 | 0.384   | 0.250  | 0.566               |
> | GMM-Synth (Ours)   | 0.227 | 0.339   | 0.238  | 0.508               |
>
> |                    | 1pp   | Hedging | Stance | CBP |
> |--------------------|-------|---------|--------|---------------------|
> | Human-Only         | 0.301 | 0.384   | 0.250  | 0.566               |
> | GMM-Synth (Ours)   | 0.198 | 0.329   | 0.229  | 0.506               |
>
> For 50% of the samples, we default to complete random noise.
>
> |                    | 1pp   | Hedging | Stance | CBP |
> |--------------------|-------|---------|--------|---------------------|
> | Human-Only         | 0.301 | 0.384   | 0.250  | 0.566               |
> | GMM-Synth (Ours)   | 0.300 | 0.374   | 0.248  | 0.567               |
>
>
> We also report the accuracy of GPT-4o, Llama-3-8b, Qwen-3-8b for these tasks to more comprehensively understand performance.
>
> |                                           | GPT-4o | Llama-3-8b | Qwen-3-8b |
> |-------------------------------------------|--------|------------|-----------|
> | Hedging (Covariate Prediction) | 0.764  | 0.485      | 0.501     |
> | Hedging (Label)     | 0.785  | 0.150      | 0.230     |
> | 1pp (Covariate)     | 0.983  | 0.878      | 0.866     |
> | 1pp (Label)         | 0.785  | 0.150      | 0.230     |
> | Stance (Covariate)             | 0.864  | 0.620      | 0.645     |
> | Stance (Label)                 | 0.742  | 0.636      | 0.621     |
> | CBP (Covariate)| 0.125  | 0.087      | 0.098     |
> | CBP (Label)    | 0.827  | 0.558      | 0.557     |
>
>
> > Tasks are simple
>
> We would like to remark that we run our experiments on standard benchmarks used in recent “LLM for statistical inference” works [3][4]. In fact, the inference tasks we use are actually more elaborate than previous work in this area, since we include a  nonlinear task (logistic regression) as opposed to more simple linear inference tasks (such as simple mean estimation, quantile estimation, etc.) used in past works [1][2]. To further address your concern, we have also included new results on ordinary least squares (OLS) – another common statistical inference task— in the following Table. We see that even for linear tasks and even when Y is not binary but continuous, the same conclusions hold. OLS and logistic regression are perhaps the most widely used statistical models in computational social science practice and strong performance on these tasks is widely applicable.
>
> |                    | 1pp   | Hedging | Stance | CBP |
> |--------------------|-------|---------|--------|---------------------|
> | Human-Only         | 0.022 | 0.021   | 0.036  | 0.034               |
> | PPI++Synth         | 0.020 | 0.022   | 0.035  | 0.026               |
> | GMM-Synth (Ours)   | 0.009 | 0.015   | 0.022  | 0.015               |
>
>
> ### **Organization of figures**
>
> > Key observations in Experiments should be structured better. Right now, jumps from Figures 3 and 4, to Figure 6 in the Appendix, to finally get to Figure 2, but then going back to Figure 3.
>
> Thank you for this suggestion! We have grouped Figure 3, 4 and Figure 6 together. Followed by Figure 2, for better flow.
>
> ### **Questions**
>
> > In L.135, do you mean $\tilde{X}_k = f( \tilde{T}_k)$ idem ditto for $Y$?
>
> Thanks for raising this clarification point! Yes, that is exactly what we mean. We have added this to the draft for better clarity.
>
> > If we have enough texts , what is the intuition for why modelling synthetic  helps?
>
> Thank you for this great question! Even when the practitioner possesses a large collection of raw texts, the bottleneck for inference is the labels $(X,Y)$, because hand-labeling texts is expensive. Intuitively, prompting the LLM to generate a synthetic sample $(\tilde{T}, \tilde{X}, \tilde{Y})$ for every real document $T$ can be viewed as obtaining a second noisy view of the same latent signal.
>
> To further answer your question, we plot the relative improvement in variance (Variance_human_only - Variance_with_synthetic) / Variance_human_only, where the number of raw text samples (real T) grows. This metric represents the percentage reduction in variance achieved by adding synthetic data. We fix the labeled portion of the raw text samples (real T) to be fixed p=0.05. We run this on the Congressional Bills dataset, as it has the largest number of raw text samples. We observe that the relative improvement is almost constant, indicating that the improvement from adding synthetic data persists even when the number of real text grows large.
>
> |                    | n=1000 | n=2000 | n=3000 | n=4000 | n=5000 |
> |--------------------|--------|--------|--------|--------|--------|
> | Relative Variance  | 0.220  | 0.210  | 0.208  | 0.211  | 0.210  |
>
> We thank the reviewer again for their time and consideration! We are more than happy to address any other concerns they might have.
>
>    1. Angelopoulos et. al. PPI++: Efficient Prediction Powered Inference.
>    2. Angelopoulos et. al. Prediction-Powered Inference.
>    3. Gligoric et. al. Can Unconfident LLM Annotations Be Used for Confident Conclusions.
>    4. Egami et. al. Using Imperfect Surrogates for Downstream Inference.
>    5. Ji et. al. Predictions as Surrogates: Revisiting Surrogate Outcomes in the Age of AI.

---

> > ### Comment · Reviewer_7qA1 · 2025-08-05
> >
> > Many thanks for the answers and additional results.
> >
> > W1) My concern was that the method details seem key to the method's performance. Thanks a lot for the additional experiments with Llama and Qwen, which show this too: the method is of course only as good as the LLM. In practice, many people prefer using Llama or Qwen over GPT-o, so I hope the authors will include these results in the final paper.
> >
> > Q2) Considering my earlier question:
> > > If we have enough texts , what is the intuition for why modelling synthetic helps?
> >
> > My point was: considering you assume you have enough texts, just not the labels, why would you generate synthetic texts, and not just synthetic labels based on the real texts you have? I don't see the benefit of introducing potential noise by making a synthetic "copy".

---

> ### Author Response · Authors · 2025-08-05
>
> Dear Reviewer 7qA1,
>
> We thank the reviewer for their appreciation of the new results. And yes, we will definitely include those in the final paper.
>
> Regarding Q2, great point! The reviewer is correct that there is less motivation for employing synthetic data if the practitioner already has a large, sufficient number of real texts for the downstream inference task. We would like to clarify that the main motivation for this work is settings where this is not the case—where practitioners have an insufficient amount of real texts for the downstream inference task (due to reasons such as budget or accessibility). We have explicitly clarified this in Section 1 (Lines 27-34) with the following clarifications (changes are bolded):
>
> """
> In these settings, where manually labeling large text corpora can be prohibitively expensive, practitioners **have leveraged** large language models (LLMs) as cheap but noisy labelers [Egami et al., 2023, Gligoric ́ et al., 2024].
>
> More recently, **in settings where real texts are insufficient**, practitioners have started to explore the possibility of **also** leveraging LLMs **not only as labelers but also** to output entirely new synthetic samples, e.g., simulating human responses to surveys or human participants in early pilot studies [Argyle et al., 2023, Brand et al., 2023, Dominguez-Olmedo et al., 2024, Anthis et al., 2025, Hwang et al., 2025b]. Such synthetic simulations aim to overcome the financial limitations and time costs of **collecting sufficient real text samples**, to improve both efficiency and accessibility [Alemayehu et al., 2018].
> """
>
> We thank the reviewer again for their appreciation of the new results. We hope these revisions and additional results address Reviewer 7qA1’s concerns. We sincerely thank the reviewer again for their valuable time and consideration and are more than happy to address any other concerns they might have.

---

> ### Author Response · Authors · 2025-08-07
> **Hope to hear back soon**
>
> Dear Reviewer 7qA1,
>
> We thank the reviewer for their appreciation of the new results. In our response above, we have tried to address all your questions and comments. To summarize, we have:
>
> 1. Added results with more baselines.
> 2. Strengthened empirical evaluations to include ablations with worse quality synthetic data: We have added results using worse quality LLMs: small, open-source models (Llama3-8b and Qwen-3-8b) and have demonstrated the same conclusions hold. We have also added an experiment where we default to complete random noise for 50% of the samples to further demonstrate a setting where the synthetic data is not reflective of the underlying task at hand and reflect predictions of a very poor model.
> 3. Added new results on OLS tasks—another common statistical inference task in the literature.
> 4. Added clarifications of method details and organization of figures.
>
> We once again thank you for your valuable time spent reviewing our work. We hope this response addresses your concerns. If so, we would be extremely grateful if you would consider updating your score. If you have any more questions or feedback, please let us know.

---

> > ### Comment · Reviewer_7qA1 · 2025-08-08
> >
> > Thank you for the last clarification. I very much appreciate the thoughtful responses and additional experiments, but I will retain my original score.

---

> ### Author Response · Authors · 2025-08-08
>
> We thank the reviewer for their appreciation of our responses and new results. We sincerely appreciate your valuable time and feedback. Best wishes

---

### Note · Authors · 2025-08-11

Dear AC and Reviewers,

Thank you for your valuable time and feedback! We're glad that our rebuttal addressed reviewers' concerns, with Reviewer B76K revising their score upward and Reviewers psyf and ​​7qA1 confirming that our responses and additional results have addressed their questions and concerns.

We were unfortunately unable to reach Reviewer WjYv during the discussion period. We thank the reviewer for their comments that they find our “contributions to be significant” and that our “empirical evaluations support our theoretical results”. To summarize, we have added the following revisions to explicitly address their concerns:

- Added a section to the Related Work (after Line 104) to provide further context on GMM earlier on in the draft. (The paragraph is explicitly provided in the rebuttal).
- Pointers to where we detail the main contribution/method and background on GMM methodology in Section 4: We introduce the moment conditions in Section 4.1; step through the estimator construction with proxy and synthetic data in Section 4.2; provide concrete examples of this in Appendix B; outline the two-step estimation procedure in Section 4.3; and provide theoretical results on the consistency and asymptotic behavior of our GMM estimator in Section 4.4 with full proofs in Appendix A.
- Pointers to our contributions outlined and summarized in lines 71-77.
- Moved up concrete examples from Appendix B into the main text to provide further clarity for readers.
- Updated the Introduction (Lines 61-70) with the additional paragraph (provided in the rebuttal) to summarize and further expand on how the GMM works and helps with the problem at hand earlier in the draft, as suggested.

We hope these revisions address Reviewer WjYv's concerns and improve our manuscript to meet their expectations for an acceptable article.  We once again thank all Reviewers and the AC for their valuable time and consideration. Best wishes.

---

### Decision · Program_Chairs · 2025-09-17

**Decision:**

Accept (poster)

**Comment:**

As I have weighed the reviewer's comments, the author's rebuttal, and my read of the paper itself, I have concluded that this paper has more potential impact that its ratings might suggest.  I believe that the problem setup and proposed solution could have application far beyond the CSS settings discussed here, and could provide important foundational ideas to increase our ability to harness synthetic data. To me, that is a critical capability that will become increasingly important in the future, and I believe now is a good time to publish papers to spark conversations and future work in this area.

I think the idea of using GMMs is mathematically elegant and well-justified (the method in this paper reminds me of control variates from control theory).  The core principle seems to be that synthetic data can be used as long as there is some sort of connection/signal/pattern between the real and synthetic data; in this paper, that signal is "covariance", but I could imagine building more sophisticated linkages.

I encourage the authors to respond to the reviewers comments and address their concerns, but especially after the additional work the authors have done during the rebuttal period, I find the paper worthy of publication.